# Isotropic reconstruction for electron tomography with deep learning

Yun-Tao Liu[1,2,3,9], Heng Zhang[1,4,9], Hui Wang[2,3,5], Chang-Lu Tao[1,6,7], Guo-Qiang Bi ✪ [1,6,8] ✉ & Z. Hong Zhou ✪ [2,3,5] ✉

Cryogenic electron tomography (cryoET) allows visualization of cellular structures in situ. However, anisotropic resolution arising from the intrinsic "missing-wedge" problem has presented major challenges in visualization and interpretation of tomograms. Here, we have developed IsoNet, a deep learning-based software package that iteratively reconstructs the missing-wedge information and increases signal-to-noise ratio, using the knowledge learned from raw tomograms. Without the need for sub-tomogram averaging, IsoNet generates tomograms with significantly reduced resolution anisotropy. Applications of IsoNet to three representative types of cryoET data demonstrate greatly improved structural interpretability: resolving lattice defects in immature HIV particles, establishing architecture of the paraflagellar rod in Eukaryotic flagella, and identifying heptagon-containing clathrin cages inside a neuronal synapse of cultured cells. Therefore, by overcoming two fundamental limitations of cryoET, IsoNet enables functional interpretation of cellular tomograms without sub-tomogram averaging. Its application to high-resolution cellular tomograms should also help identify differently oriented complexes of the same kind for sub-tomogram averaging.

The advent of single-particle cryoEM has made it routine to determine structures of isolated macromolecular complexes at 2–4 Å resolution by averaging hundreds of thousands of particles, enabling atomic modeling. The biological functions of these complexes, however, are carried out through their interactions and often depend on their spatial arrangements within cells or sub-cellular organelles[1,2]. Examples abound, ranging from pleomorphic viruses[3] to cellular organelles[4], to large-scale cellular structures like synapses between neurons[5,6]. Many viruses, notably those involved in devastating pandemics such as

SARS-CoV-2[7–9], influenza viruses[10], and human immunodeficiency viruses (HIV)[11], are pleomorphic in the organizations of their proteins and genomes. Cellular organelles, such as axonemes containing microtubule doublets surrounding a central pair[12], though largely conserved in their core elements across different species, rely on their non-conserved and variable attachment of peripheral components that define their characteristic species-specific functions[13]. In neurons, organizations of molecules, rather than molecules alone, inside the synapse might underlie synaptic plasticity that is generally regarded as

[1]Center for Integrative Imaging, Hefei National Research Center for Physical Sciences at the Microscale, School of Life Sciences, Division of Life Sciences and Medicine, University of Science and Technology of China, Hefei, Anhui 230026, China. [2]California NanoSystems Institute, University of California, Los Angeles (UCLA), Los Angeles, CA 90095, USA. [3]Department of Microbiology, Immunology and Molecular Genetics, UCLA, Los Angeles, CA 90095, USA. [4]Department of Physics, University of Science and Technology of China, Hefei, Anhui 230026, China. [5]Department of Bioengineering, UCLA, Los Angeles, CA 90095, USA. [6]Interdisciplinary Center for Brain Information, Brain Cognition and Brain Disease Institute, Shenzhen-Hong Kong Institute of Brain Science-Shenzhen Fundamental Research Institutions, Shenzhen Institute of Advanced Technology, Chinese Academy of Sciences, Shenzhen 518055, China. [7]Faculty of Life and Health Sciences, Shenzhen Institute of Advanced Technology, Chinese Academy of Sciences, Shenzhen 518055, China. [8]Center for Excellence in Brain Science and Intelligence Technology, Chinese Academy of Sciences, Shanghai 200031, China. [9]These authors contributed equally Yun-Tao Liu, Heng Zhang. ✉ e-mail: gqbi@ustc.edu.cn; Hong.Zhou@UCLA.edu

the cellular basis of learning and memory[6,14]. Such organizational information, or "molecular sociology", unfortunately, is lost in single-particle cryoEM.

To reveal such molecular sociology across viruses or inside cells, cryogenic electron tomography (cryoET) has become the tool of choice[15,16]. This technique requires collecting a series of images at different tilt angles, called a "tilt series". Due to radiation damage, limited electron dosage must be further fractionated throughout the tilt series, resulting in a low signal-to-noise ratio (SNR) in the cryo tomogram. Furthermore, as tilting increases the effective thickness of the sample, the tilt range for cryoET is usually restricted to about ±60°. The missing views at higher tilt angles result in anisotropic resolution of the reconstructed 3D tomograms, with the lowest resolution along the Z-axis (Supplementary Fig. 1). In Fourier space, these missing views lead to a devoid of information in two continuous, opposing wedge-shaped regions, commonly referred to as the "missing-wedge". This missing-wedge causes severe artifacts in the 3D reconstruction of cellular cryoET, manifesting as, e.g., oval-shaped synaptic vesicles[17] (Supplementary Fig. 1). Thus, together with the low SNR in the reconstructed tomograms, the presence of missing-wedge artifacts prohibits direct interpretation of the reconstructed densities in 3D, which is key to the promise of cryoET to resolve molecule organization in situ.

Previous attempts have been made to partially recover information in the missing-wedge[18–20] with a priori assumptions (e.g., density positivity and solvent flatness, both of which are binary) to constrain the structural features in reconstructed tomograms. However, such binary assumptions have limited information content (or "entropy") and may not always hold true, given the complexity of biological systems (for example, certain regions of cellular space may contain non-membrane-bounded phases of matters other than a flat solvent). Alternatively, dual-axis tomography relies on imaging the same sample with two perpendicular tilt axes, reducing the two missing-wedges to two missing pyramids; thus, it has the potential to alleviate artifacts in resulting tomograms[21]. However, the acquisition and alignment of dual-axis tilt series are substantially more complicated than that of single-axis tilt series and could waste the already limited electron dose used for tilt series aquisition[22]. Consequently, while implemented in high-end instruments such as the Thermo Fisher Titan Krios, dual-axis tomography has not been practically attractive. Indeed, to date, no structure with better than nanometer resolution was obtained from dual-axis tomography.

Deep neural networks are known to learn relationships of complex data that are non-linear or have high dimensionality. In computer vision, convolutional neural network (CNN) has been applied to various tasks, such as object recognition[23], image segmentation[24], and classification[25], often achieving high performance. In the cryoEM community, CNN-based neural networks are applied to particle-picking tasks and outperform conventional methods such as the Laplacian of Gaussian approach[26]. Deep learning was also applied to cryoET tomograms for denoising[27–29], particle picking[30,31], classification[32,33], and segmentation[34,35], accelerating the interpretation and analysis of the tomograms. Among them, DeepFinder[30], and EMAN2[34] can produce segmentation masks of macromolecules without considering the missing-wedge artifacts. The deep learning-based method was also used for unsupervised subtomogram classification employing feature space learned in a supervised classification task[32]. The Noise2Noise strategy[36] (i.e., learning from one noisy instance to another noisy instance of an image) in these methods requires no noise-free ground truth and only makes assumptions on the independency and zero-mean statistics of the noise. However, although the latest EMAN2 release attempts to fill in the missing-wedge with somewhat meaningful information with cycleGAN[37], it could not recover reliable large-scale information such as flat membrane; none of the other packages dealt with the missing-wedge problem in cryoET. Consequently,

whether CNN can recover reliable information into the "missing-wedge" for cryoET reconstruction has not been explored.

Here, we have developed a CNN-based software system, called *IsoNet*, for isotropic reconstruction for electron tomography. IsoNet works for low-resolution tomograms at a pixel size of 10 Å. It trains deep CNN that iteratively restores meaningful contents to compensate for missing-wedge, using the information learned from the original tomogram. The resolution within the missing-wedge reaches about 30 Å resolution as measured by the gold-standard Fourier shell correlation (FSC) criterion. By applying IsoNet to tomograms representing viral, organelle, and cellular samples, we demonstrate its superior performance in resolving novel structures of lattice defects in immature human immune-deficiency virus capsid, the scissors-stack-network architecture of the paraflagellar rod, and heptagon-containing clathrin cage inside a neuronal synapse. The resulting tomograms with isotropic resolution from IsoNet should help direct interpretation and segmentation of 3D structure in cells and 3D picking hundreds of thousands of subtomogram particles for future high-resolution cryoET studies.

## Results
### Workflow of IsoNet
In spite of anisotropic resolution, tomograms generated by cryoET reconstruction contain rich information with structural features such as plasma membranes, organelles, and protein complexes. It is possible to recover the missing information by merging information from similar features present in the same tomograms but at different orientations relative to each other. An example of filling such missing information is through subtomogram averaging, which aligns and averages structures of particles that are identified to be identical but at different orientations in the tomogram. IsoNet is designed to expand this technique to reconstruct missing information by training the neural network targeting the subtomograms at different rotations for both regular and polymorphous structures.

The workflow of IsoNet contains five steps (Fig. 1a). Among them, three are major and required: *Extract*, *Refine*, and *Predict*; and the other two are optional: *Deconvolve CTF* and *Generate Mask*. Each of these steps can be performed with one command of IsoNet in a Linux terminal. Among the five steps, *Refine* and *Predict* requires GPU acceleration. The input of IsoNet is either from a single or multiple tomograms. More tomograms will generate more reliable results but take a longer processing time. The typical number of tomograms for IsoNet is from one to five in practice. The tomogram(s) can be reconstructed by either weighted backprojection (WBP) or iterative methods, such as the simultaneous iterative reconstructive technique (SIRT).

We implemented IsoNet in Python using Linux as the native operating system. The package takes advantage of the Keras interface of the well-established Tensorflow platform[38]. The package can be run either through the command line or through a graphical user interface (GUI) (Fig. 1b), thus meeting the needs of both seasoned and novice cryoET investigators. The GUI contains three tabs to facilitate navigation. In each tab, tomogram information and the parameters for each command can be specified. By clicking "*Deconvolve*", "*Generate Mask*", "*Extract*", "*Refine*", and "*Predict*" buttons, the user can execute the corresponding command. The "only print command" option prints out the corresponding command for each step which can be executed on other computers or submitted to computer clusters.

*Deconvolve CTF* and *Generate Mask* steps are two optional steps performed on the input tomograms prior to the subtomogram extraction in the *Extract* step (Fig. 1a). The *Deconvolve CTF* step has two purposes: enhancing low-resolution information and compensating for the contrast transfer function (CTF) in the tomograms acquired at certain underfocus conditions. Due to the presence of zeros in CTF, we used a Wiener filter for CTF compensation, as implemented in Warp[28].

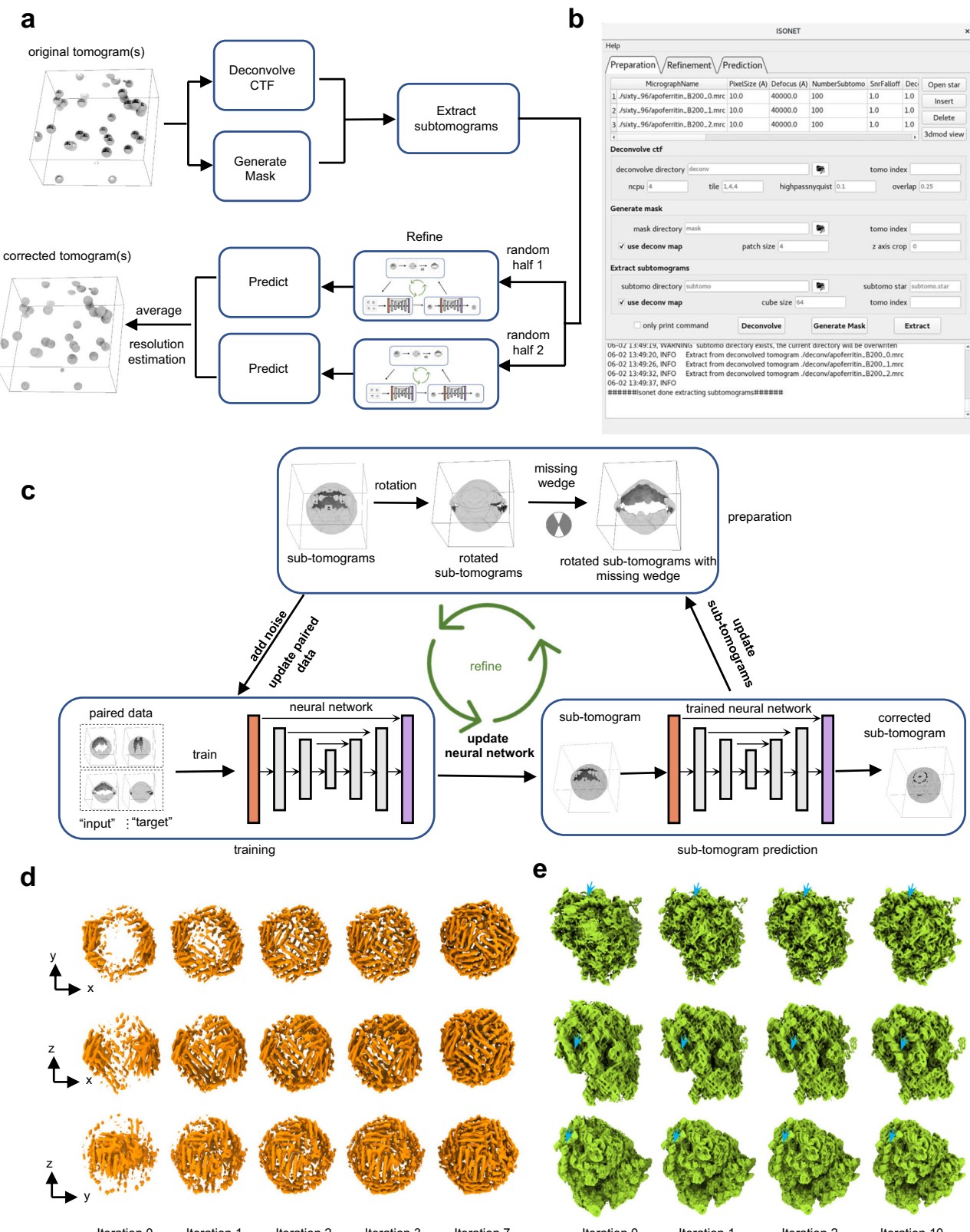

**Fig. 1 | Principle and workflow of IsoNet. a** Workflow of the IsoNet software. **b** GUI of IsoNet. **c** Illustration of *Refine* step: First, subtomograms are rotated and then applied with additional missing-wedge artifacts along other directions (e.g., YZ axis) to produce paired data for training. Second, the paired data is used to train a neural network with U-net architecture. Third, the trained neural network is applied to the extracted subtomograms to produce missing-wedge corrected subtomograms. The recovered information in these subtomograms is added to the original subtomograms, producing new datasets for the next iteration. **d**, **e** Validation of IsoNet with simulated subtomogram of apoferritin (**d**) and ribosome (**e**). Surface views from three orthogonal directions of the reconstructions are shown after increasing iterations of IsoNet processing. Blue arrows indicate segments of RNA duplexes.

The *Generate Mask* step uses statistical methods to detect "empty" areas in the tomograms (including vacuum above and below the sample and those only containing ice or carbon) to be excluded from the subsequent analysis. Both steps could improve the performance and efficiency of neural network training.

The *Extract* step allows randomly cropping subtomograms in the original tomograms or the region of interest of the tomograms defined by masks. The maximum sizes of subtomograms depend on the memory of graphics processing units (GPU), and $64^3$ or $96^3$ voxels are often used. The extracted subtomograms can be split into random halves to train the neural network independently (Fig. 1a), allowing users to perform 3D gold-standard FSC[39,40] to determine the resolution of IsoNet reconstructed tomograms over different angular directions, particularly in the Z-axis.

Central to the IsoNet workflow is the *Refine* step, which iteratively trains neural networks to perform missing-wedge correction and denoising (Fig. 1c). Training the neural network requires paired subtomograms as the "inputs" and the "targets." The "targets" for IsoNet are the extracted subtomograms rotated at different orientations. In total, 20 different orientations are defined in IsoNet, generating 20 "target" subtomograms for each extracted subtomogram (Supplementary Fig. 2). For each "target" subtomogram, an additional missing-wedge is computationally imposed in Fourier space to generate the corresponding "input" subtomograms (Fig. 1c). The direction of this additional missing-wedge is different from the missing-wedge in unrotated tomograms. After generating the paired dataset, we train a neural network to map the "input" to the "target", enabling the network to recover the imposed missing-wedge artifacts. The IsoNet adopts U-Net architecture[41], containing an encoder path that extracts low-dimensional representation retaining essential properties, a decoder path to reconstruct from the encoded representation, and skip-connections between the encoder and decoder to preserve high-resolution information (Supplementary Fig. 3).

However, the "target" in the data pairs described above are not ideal subtomograms. These subtomograms, though rotated, still miss information in other directions. To recover that information and make "target" subtomograms resembling "ground truth", we adopt an iterative approach: In the first iteration, we train the network with subtomograms generated from the *Extract* step and obtain the IsoNet-predicted subtomograms. Then, the gained information in the missing-wedge region in the Fourier space of the predicted subtomograms was added to the original subtomograms, generating the first-iteration missing-wedge corrected subtomograms (Fig. 1c). To further improve missing-wedge correction with more iterations, the corrected subtomograms from the previous iteration are used for the paired data generation in the next iteration because they are closer to missing-wedge-free 3D volumes than the extracted original subtomograms. The trained network from the previous iteration is then refined with the newly generated data pairs. Through multiple iterations, the missing-wedge information is gradually added to the subtomograms (Fig. 1c and Supplementary Fig. 4). Usually, after 10–20 iterations, the refinement converges when the mean square error no longer decreases. With a dataset of 150 subtomograms, the *Refine* step typically takes about 10 h for four Nvidia-1080Ti GPUs.

To prevent error accumulation in the iterative refinement, the predicted subtomograms in previous iterations are not directly used for the next iteration. Instead, the subtomograms are prepared by adding the missing-wedge region of corrected subtomograms to the original subtomograms. Any information from the original subtomograms will not be modified throughout the iterative process, ensuring the fidelity of the refinement. Therefore, we have not observed any error accumulation during the *Refine* step.

Within the *Refine* step of IsoNet, we also implemented a denoising module based on the noisier-input strategy[42,43]. When this optional denoising module is enabled, in each iteration, 3D noise volumes are reconstructed by the backprojection algorithm from a series of 2D images containing only Gaussian noise. Those 3D noise volumes are added to "input" subtomograms, with the "target" subtomograms staying the same. With this strategy, the neural networks can be robustly trained with these noisier "input" subtomograms to eliminate the added noise and improve the SNR of final isotropic reconstructions (Fig. 1c and Supplementary Fig. 4).

IsoNet embodies several measures that prevent the neural network from "inventing" molecule features. First, the neural network was initialized with random numbers, and all the information came from original tomograms without prior knowledge. Second, we introduced the dropout factor of 0.5 in the neural network so that with 50% of randomly picked neurons remaining, the network can still reproduce the result. Third, the extracted subtomograms for training can be divided into random halves, and the resolution estimation is based on the gold-standard 3D FSC. Another way to alleviate overfitting is adding a more diverse dataset for network training, although time consumption increases proportionally to the number of subtomograms.

The result of this *Refine* step is a trained network that will be applied to the full tomograms and produce the isotropic reconstruction in the *Predict* step (Fig. 1a). Tomograms used for *Predict* step are typically (preferably because there are no concerns of bias) the same or a subset of the tomograms used to train the network. Nonetheless, users can apply the trained network to tomograms of other similar samples. The time consumption in the *Predict* step for one tomogram of $1000 \times 1000 \times 300$ pixels is about 5 min using four Nvidia-1080Ti GPUs.

## Benchmarking with simulated data

We first performed IsoNet reconstruction on simulated subtomograms using publicly available atomic models. Two scenarios have been considered: apoferritin[44] for the first test because it has been widely used as a benchmarking specimen in high-resolution cryoEM and ribosome[45] as the second test due to its asymmetric shape. For both tests, density maps were simulated from the atomic models using *molmap* function in Chimera[46] and filtered to 8 Å resolution (Fig. 1d, e). The simulated maps were then rotated in ten random directions and imposed missing-wedge in Fourier space, resulting in simulated subtomograms with missing-wedge artifacts (leftmost columns in both Fig. 1d, e).

As evident in both tests with simulated subtomograms, features such as alpha helices perpendicular to the Z-direction are smeared out in those simulated subtomograms due to the missing-wedge artifact. IsoNet was then used to process those simulated subtomograms. As expected, the missing information was recovered during this iterative refinement process (Fig. 1d, e). After seven iterations, all the alpha helices are visible and identical to the ground truth structures in the first test. The cubic symmetry of apoferritin gradually emerged even though we did not impose symmetry during IsoNet processing. The distortion in the shapes of ribosomes is reduced during the *Refine* step, with the major and minor grooves of the RNA becoming distinguishable (Fig. 1e). These results indicate that IsoNet performed well with simulated round/symmetric protein complex as well as an asymmetric complex containing both protein and nucleic acid.

## Application to ribosome and virus tomograms

To further demonstrate the superior performance of IsoNet in real-world examples, we performed the IsoNet reconstruction with two well-characterized cryoET datasets available from the Electron Microscopy Pilot Image Archive[47], including ribosome[48] (EMPIAR-10045), and virus-like particles (VLP) of immature HIV-1[11] (EMPIAR-10164).

For the ribosome dataset, the contrast of IsoNet-processed tomograms is higher than that of CTF deconvolved tomograms (Fig. 2a, b), enabling direct segmentation of ribosomes in 3D using a single threshold (Fig. 2c, d). Characteristic ribosomal features,

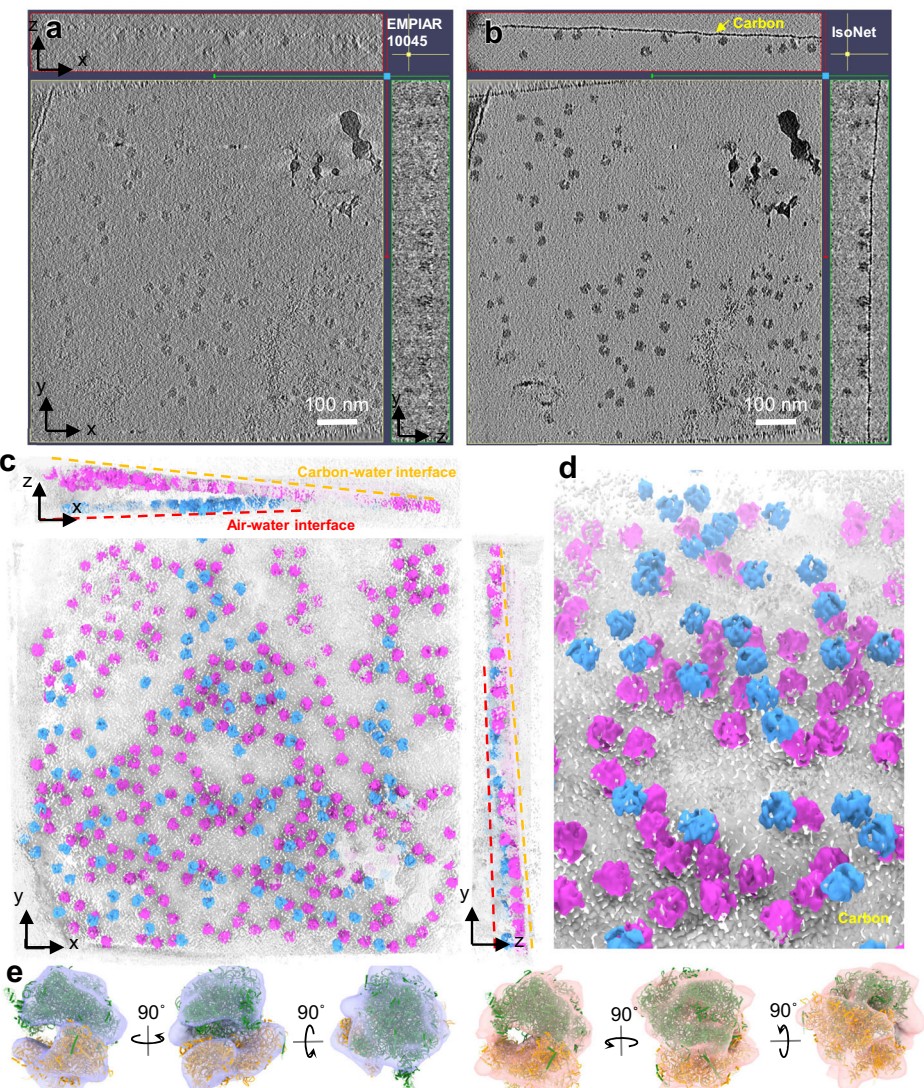

**Fig. 2 | IsoNet corrects missing-wedge artifact of ribosome dataset.**
**a**, **b** Orthogonal slices of original (**a**) and IsoNet-processed (**b**) tomograms of isolated ribosomes. **c** 3D rendering of the missing-wedge corrected tomogram. Dashed lines show the air-water or carbon-water interfaces. **d** Enlarged view of the tomogram showing the structures of individual ribosome proteins. **e** Densities of ribosomes in IsoNet-processed tomograms (transparent surface) fitted with atomic models of the ribosome. Green ribbons: large ribosomal subunits; Orange ribbons: small ribosomal subunits.

including large and small ribosomal subunits, can be readily distinguishable in 3D. The atomic structures of the ribosomes fit well into the IsoNet-processed tomograms (Fig. 2e). Remarkably, a carbon layer can be readily visible in both the XZ and YZ planes of the IsoNet-processed tomogram but not in the original tomograms. We observed that most ribosomes are attached to this carbon-water interface, and the rest are attached to an air-water interface (Fig. 2c). Distinguishing carbon-water and air-water interfaces is important because particles at the latter interface are typically damaged and should not be included in sub-tomographic averaging.

We then processed HIV tomograms with IsoNet. The gold beads in the IsoNet corrected HIV tomograms appear spherical (Fig. 3a), as they should, instead of the "X" shape due to the missing-wedge problem. Notably, the top and the bottom of the HIV particles can now be observed in the IsoNet-generated tomogram. When examined in the Fourier space, the missing-wedge region on the XZ slices was filled with values compared to the Fourier transform of the original tomogram (Fig. 3a). To quantify the resolution of the filled information, we spilt the extract subtomograms into two random subsets, trained two neural networks using those two subsets independently, and then

performed 3D FSC calculation[39]. The resolution on the XY plane is higher than other planes (Fig. 3b), with the resolution along the X- and Y-axis reaching the Nyquist resolution, showing our network preserves the high-resolution information of the original tomograms. The Z-axis resolution of the isotropic resolution is about 30 Å (Fig. 3b), which is the lowest resolution in all directions. This result demonstrates that our isotropic reconstruction can faithfully reconstruct the missing-wedge information at 30 Å resolution.

Importantly, our isotropic 3D reconstruction shows that the quality of the structure is similar across all directions at low resolution, allowing biological structures to be interpreted adequately (Fig. 3c and Supplementary Video 1). We resolved those broken viruses are sheared along the top and bottom planes of the tomograms (Fig. 3c and Supplementary Video 2), indicating that the air-water interfaces caused deformation of the capsid, as well-recognized in the cryoEM field[49]. The Gag proteins—subunits of the capsids, are mostly featureless at the air-water interfaces (Fig. 3c).

The spherical viruses fully embedded in ice are made of hexagonal lattices (Fig. 3c), whereas no pentagon subunit is observed, consistent with the subtomogram averaging results of immature HIV particles[11].

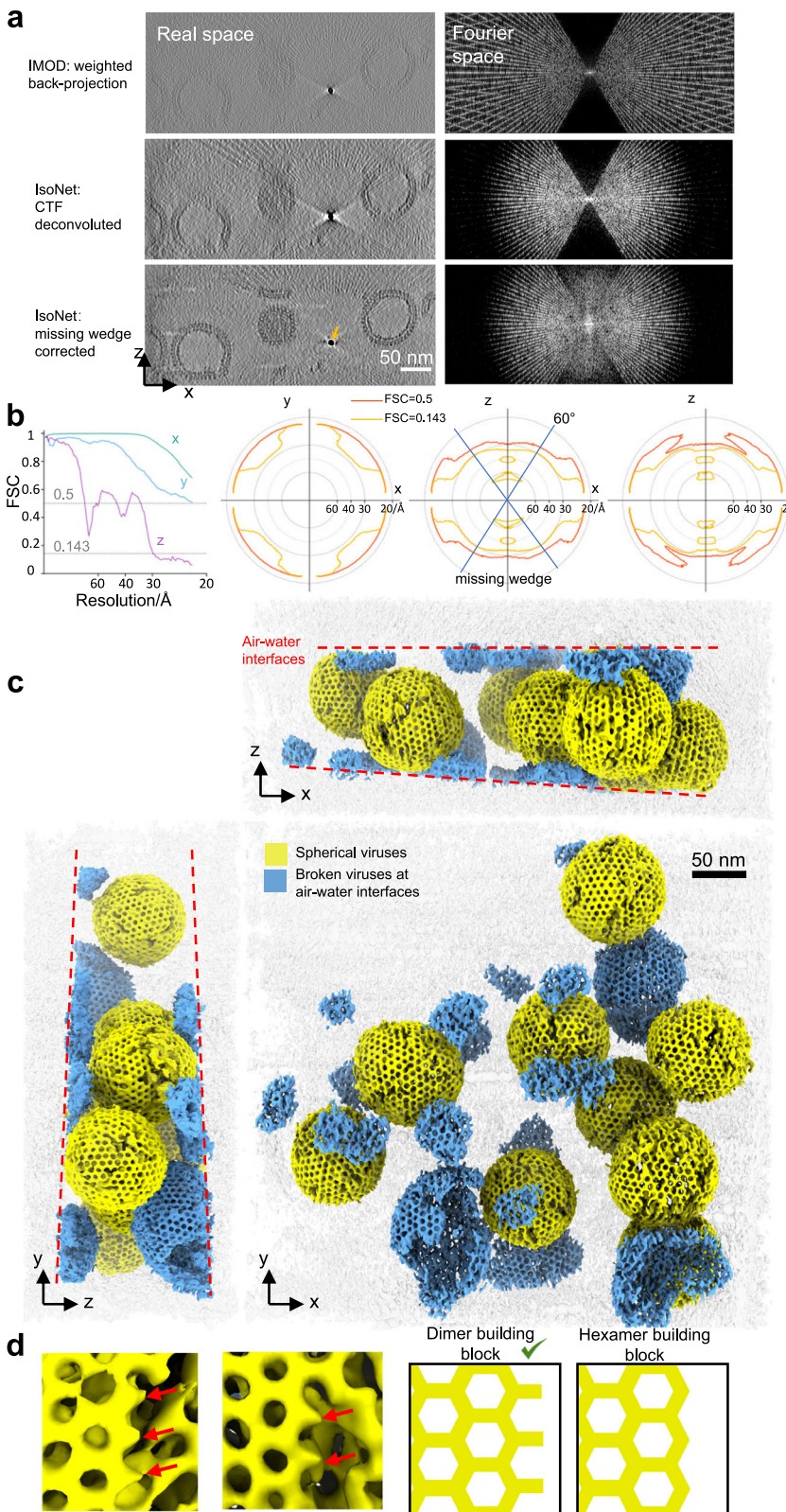

**Fig. 3 | IsoNet reveals lattice defects in immature HIV capsid. a** XZ slice views of the tomogram reconstructed with WBP (top), CTF deconvoluted WBP tomogram (middle), and missing-wedge corrected tomogram (bottom), with their Fourier transforms on the right. The orange arrow indicates a gold bead. **b** 3D FSC of the two independent isotropic reconstructions, the left panel shows the FSC along the X, Y, and Z directions. Three panels on the right show the 3D FSC at 0.5 and 0.143 cutoffs on XY, XZ, and YZ planes. **c** 3D rendering of the missing-wedge corrected tomogram. Dashed lines show the air-water interfaces. **d** Examples (left) and illustrations of the lattice edges of HIV capsids. Red arrows point out the density protrusions on the edges of hexagonal lattices.

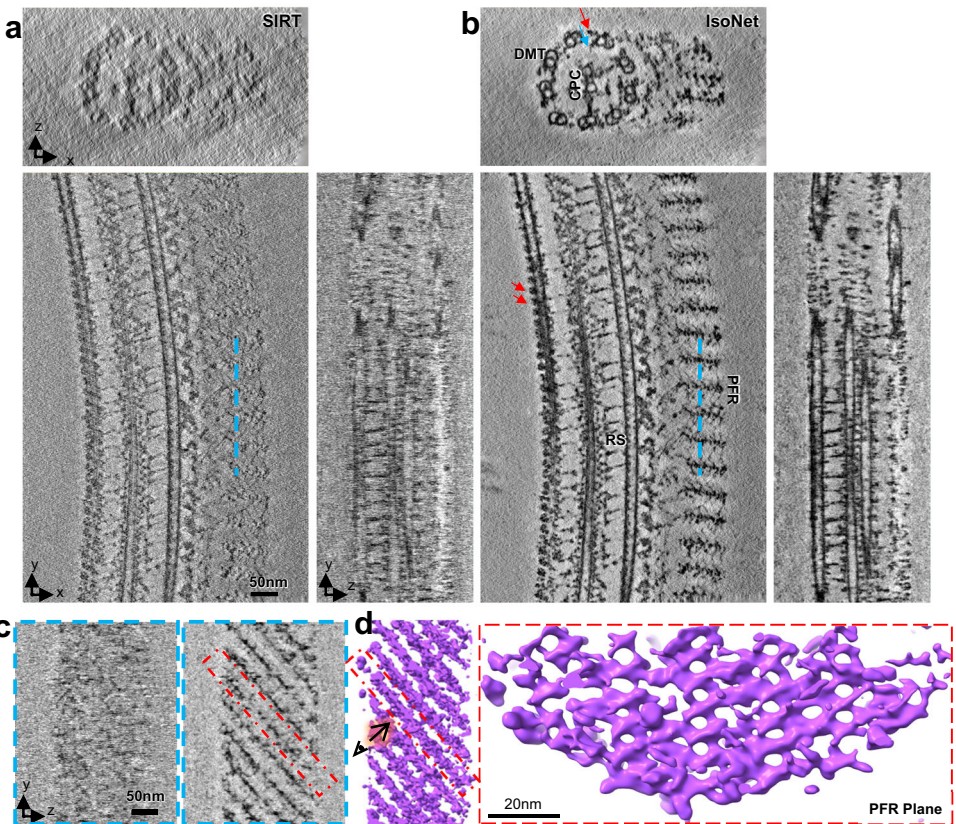

**Fig. 4 | Architecture of the PFR revealed after missing-wedge correction.**
**a**, **b** Orthogonal slices of a tomogram of flagellar for SIRT reconstruction (**a**) and IsoNet reconstruction (**b**). DMT doublet microtubule, CPC central-pair complex, RS radial spoke, Red arrows outer arm dyneins, Blue arrow inner arm dynein. **c** YZ slices show the cross-sections corresponding to the cyan lines in (**a**) and (**b**). **d** 3D rendering of PFR in the IsoNet-generated tomogram. The left panel is the 3D view of PFR in the direction corresponding to (**c**). The right panel shows the en face view of a PFR scissors-stack-network plane.

Lattice defects are incorporated into the hexagonal lattices, making gaps between patches of the lattices (Fig. 3c). These defects and slight curvature on the hexagonal lattices could enable the formation of the spherical shape without pentagons. On lattice edges, small density protrusions extending from the hexagons were observed (Fig. 3d), indicating the complete hexagons are not assembly units of HIV. In concert with this observation, a recent study shows the Gag dimers are the basic assembly units of the HIV-1 particle[50]. These protrusions could be Gag dimers and are prone to structural changes during proteolytic cleavage[50]. Those 3D details on HIV lattices can only be directly visualized after being processed by IsoNet. The above observations demonstrate that IsoNet can effectively compensate for the missing-wedge problem for relatively thin but heterogeneous structures, such as the immature HIV particles, and reach about 30 Å Z-axis resolution.

### Application to tomograms of cellular organelles
Next, we tested the performance of IsoNet on resolving structures within cellular organelles by processing tomograms of flagella of *Trypanosoma Brucei*[51] using IsoNet. The missing-wedge compensated tomogram shows relatively uniform or isotropic structures in all three dimensions (Fig. 4a, b). The overall contrast is better than the original tomogram, partially due to the denoising of the network. One noticeable missing-wedge artifact is that it is difficult to recognize the well-established 9 (outer doublets) + 2 (central-pair singlets) microtubule arrangement in the cross-section view (i.e., XZ view in Fig. 4a). This arrangement can be readily visible in the result generated by IsoNet (Fig. 4b). The missing-wedge effect is also reflected by the broken and oval-shaped microtubules and severe artifacts in XZ and YZ planes in the original tomogram reconstructed with SIRT algorithm (Fig. 4a). In tomograms generated by IsoNet, the microtubules become

complete and circular-shaped with some visible tubulin subunits (Fig. 4b and Supplementary Fig. 5). Binding to the microtubules, the arrays of outer (red arrows in Fig. 4b) and inner (blue arrows in Fig. 4b) arm dynein proteins are now clearly distinguishable in the IsoNet-generated tomogram. And radial spokes connecting the outer doublets to the central pair can be distinguished in all three orthogonal slices (Fig. 4a, b).

On one side of 9 + 2 microtubules lies a paraflagellar rod (PFR). The structure of PFR is obscure in the tomogram reconstructed by SIRT (Fig. 4c), which has given rise to the long-lasting debate on the PFR organization[52–54]. The IsoNet-generated tomograms showed a much clearer picture of PFR. PFR density consists of parallelly arranged planes, and the angle between those planes and the direction of the axoneme is 45°. Within these planes, scissors-like densities stack upon each other, forming a scissors-stack network (Fig. 4d). This highly organized mesh structure could serve as a biological spring to assist the movement of the flagella. This unique PFR structure observed here is consistent with the organization resolved through tedious efforts of subtomogram averaging thousands of subtomograms[51]. The above observations demonstrate that IsoNet can compensate for the missing-wedge problem for nonspherical cellular organelles, such as those in the Eukaryotic flagella, and unveil structure with meticulous details without the need for subtomogram averaging.

### Applications to tomograms of cells
To evaluate IsoNet's performance for much larger and more complex structures in cells, we applied IsoNet to tomograms of synapses in cultured hippocampal neurons[17]. Hippocampal synapses are key devices in brain circuits for information processing and storage. They are about 200–1000 nm in size and rich in proteins, lipid membranes,

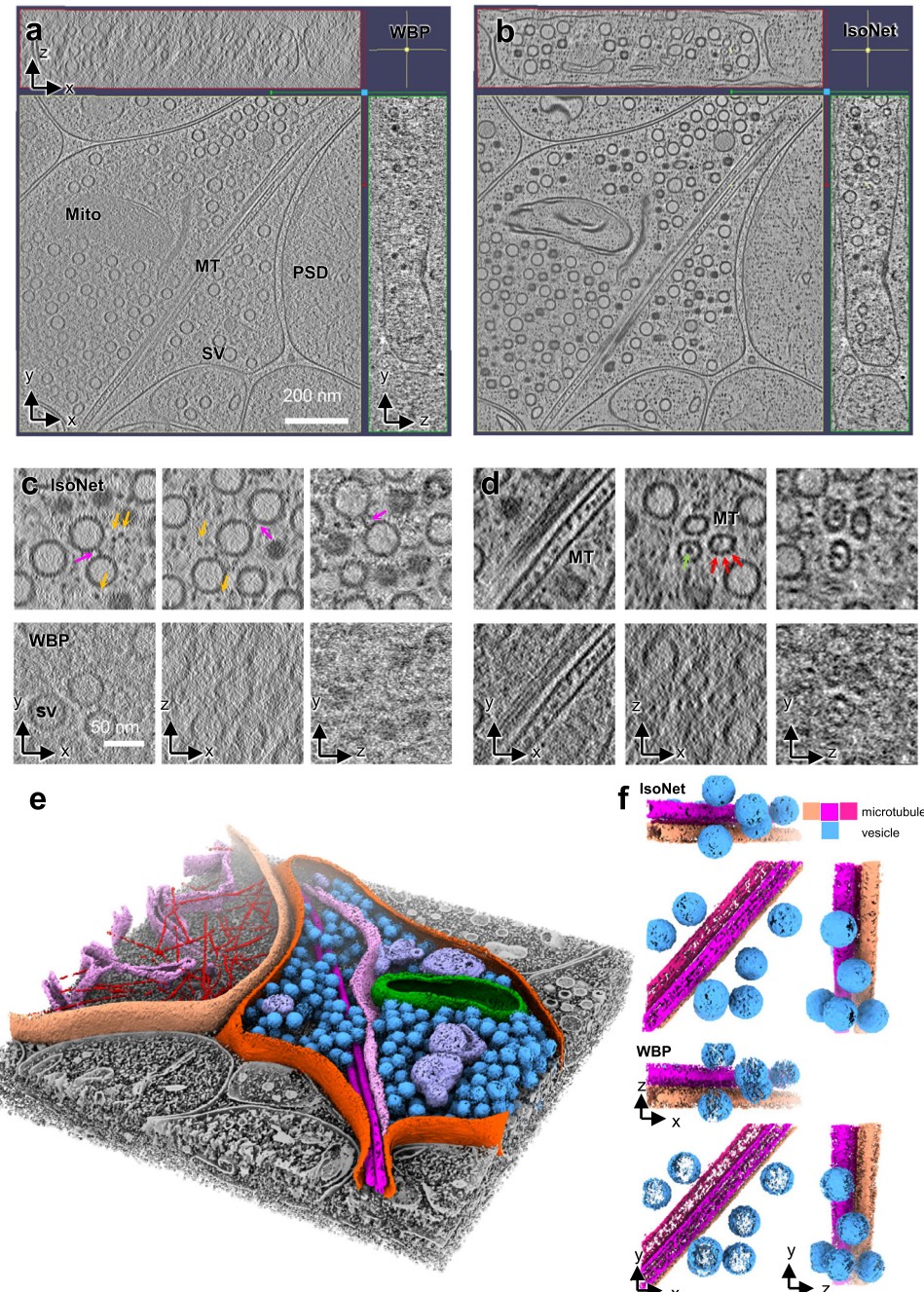

**Fig. 5 | IsoNet recovers missing information in the tomograms of neuronal synapses. a, b** Orthogonal slices of a synaptic tomogram reconstructed with WBP (**a**) and IsoNet (**b**). SV: synaptic vesicle; Mito mitochondria, MT microtubule, PSD postsynaptic density. **c, d** Zoomed-in orthogonal slices of WBP reconstruction and IsoNet-produced reconstruction. Magenta arrows: vesicle linker; Orange arrows: small cellular proteins; Green arrows: microtubule luminal particles; Red arrows: microtubule subunits. **e** 3D rendering of the tomogram shown in (**b**). **f** 3D rendering of a slab of tomogram with WBP reconstructions and Isotropic reconstructions, showing microtubules and vesicles.

vesicles, mitochondria, and other organelles[14,17,55]. These samples are thicker[17] (300–500 nm) than the above-described flagella and virus samples, thus representing low SNR tomograms. The intrinsic molecular crowding and structural complexity of the synapse also present difficulties for missing-wedge correction. Arguably, synaptic cryoET tomograms are among the most challenging datasets for any analysis algorithm. Indeed, the missing-wedge of cellular tomograms cannot be fully recovered using previous software (Supplementary Fig. 6).

IsoNet achieved isotropic reconstruction of the synaptic tomogram with substantially higher contrast and better structural integrity (Fig. 5a, b and Supplementary Videos 3–5). Synaptic vesicles that are smeared out along the Z-axis in the original tomograms now become spherical (Fig. 5c). The linker proteins between vesicles that are hardly seen in the original tomograms now become visible in XZ and YZ planes (Fig. 5c). Even some horizontally oriented features can be resolved. For example, large patches of membrane on the top and the bottom of the synapse and the endoplasmic reticulum (ER) appear smooth and continuous in the isotropic reconstruction (Fig. 5b, e). These structural integrity improvements facilitate the segmentation of the cellular structure since the missing-wedge corrected structures can be directly displayed based on their density threshold in 3D. Particularly, placing the artificial spheres to represent synaptic vesicles, as in previous

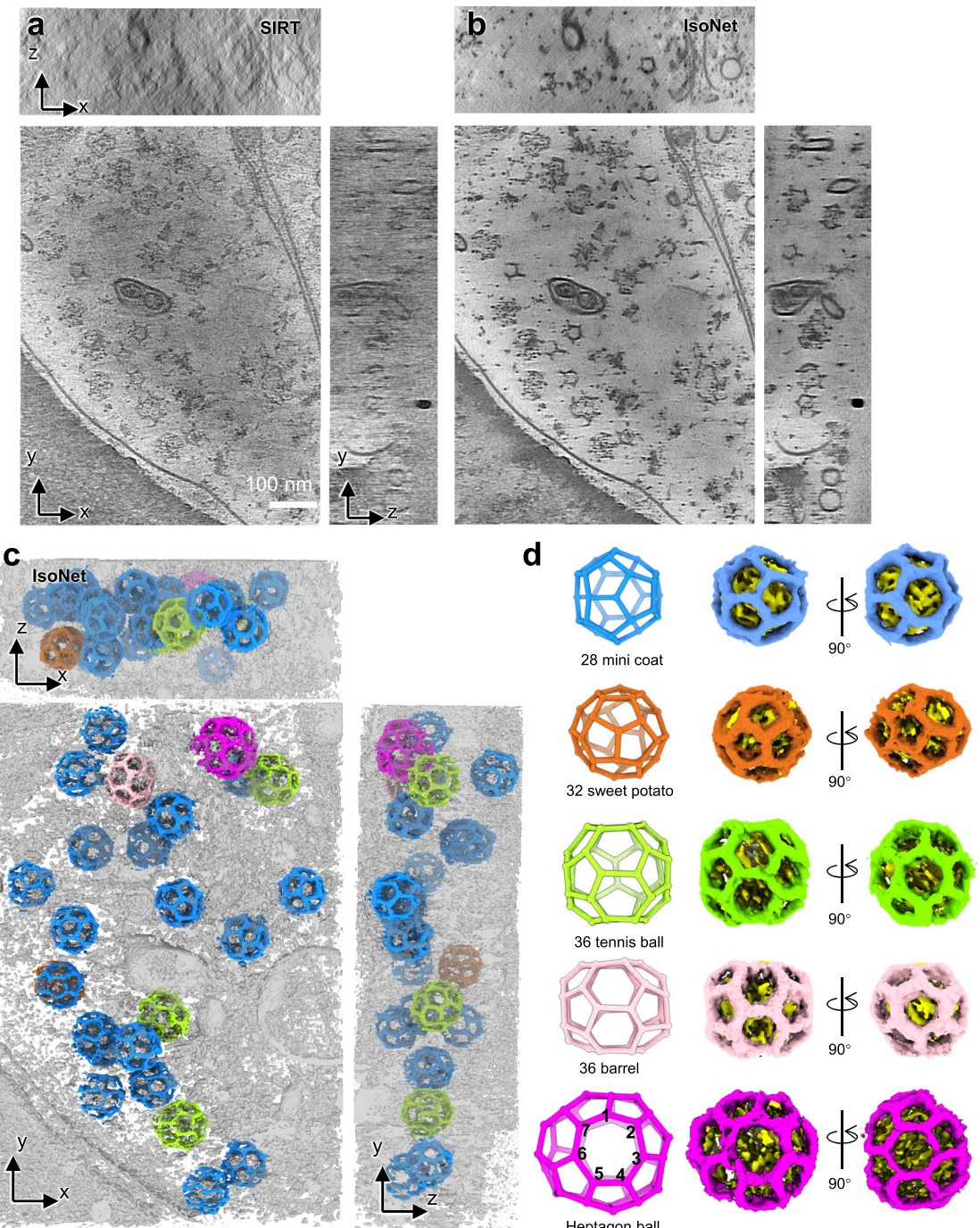

**Fig. 6 | IsoNet reveals various types of clathrin cages in a synapse.**
**a**, **b** Orthogonal slice views of another synaptic tomogram reconstructed with SIRT algorithm (**a**) and IsoNet (**b**). **c** 3D rendering of the tomogram shown in **b**. **d** 3D views of the five types of clathrin cages in **c**.

studies[17,55], is no longer needed (Fig. 5e). As the elongation effect of microtubules in the Z-axis is corrected, the protofilaments of microtubules have now become visible (Fig. 5d, f). Inside synapses, numerous small black dots can be observed in the cytoplasm but not in vesicular lumens. These dots represent small cytoplasmic proteins (orange arrows in Fig. 5c), indicating that our IsoNet preserves delicate structural features.

As a prominent example, tomograms from IsoNet revealed various types of clathrin coats in hippocampal synapses. Clathrin-mediated endocytosis is a well-known presynaptic vesicle recycling mechanism and is a critical step in synaptic transmission[56,57]. Clathrin proteins are also present in the postsynaptic compartment for neurotransmitter receptor endocytosis, a process playing essential roles in synaptic plasticity[58]. Those clathrin proteins are known to form cages that consist of pentagons and hexagons[59]. In a synaptic cryoET tomogram, we observed clathrin cages of various sizes in the postsynaptic compartment (Supplementary Fig. 7). However, due to the missing-wedge effect, the geometry of these clathrin cages cannot be directly resolved. After applying IsoNet to the tomogram, all the pentagons and hexagons, which made up the clathrin cages, are revealed (Fig. 6a–d). Whereas in the original tomograms, those polygons are smeared in XZ and YZ planes.

The 25 clathrin cages can be categorized into five types based on their geometry (Supplementary Video 6). The most abundant type is

minicoat, the smallest cages in the clathrin proteins can form[60]. Intriguingly, the largest clathrin cage contains two heptagons, in addition to 8 hexagons and 14 pentagons (Fig. 6d and Supplementary Fig. 8), which has not been reported in previous single-particle analysis[59,60]. This geometry of the cage deviates from the common belief that a closed polyhedral protein cage contains 12 polygons. This heterogeneity in the Platonic cages of the clathrin arises from the specific yet variable forms of clathrin triskelion interactions. Adapting those heptagons in neurons could likely be a strategy to scale up the size of the clathrin coats that enables accommodating different sizes of vesicles. Intriguingly, we did not observe vesicles inside these clathrin cages, suggesting that clathrin protein molecules may spontaneously self-assemble into cages even when not involved in the endocytosis. It is important to note that the unexpected heptagon-containing clathrin cage would be lost in averaging-based methods because it only has a single instance in the tomogram.

The above observations made in neurons demonstrated that IsoNet enables compensating for missing-wedge for structures that are highly heterogeneous, with limited copy numbers, and in the complex cellular environment. Because of the low SNR of the cellular tomograms, a few parameters have to be considered in IsoNet processing. First, IsoNet models missing-wedge from −60° to 60°, without tilt angle on the X-axis. Thus, application to tomograms from tilt series spanning less than 60° may not produce the ideal result. Second, binning the tomograms and using larger subtomograms increase the absolute size of the receptive field of artificial neural networks, which is important for recovering large-scale features such as cell membranes. Third, IsoNet works better for high-contrast tomograms than low-contrast ones. One efficient way to enhance contrast is to perform CTF deconvolution and tune the *snrfalloff* value. By correctly adjusting the above parameters, we are able to routinely correct missing-wedge artifacts in the tomograms for thick (300–400 nm) cellular samples (Fig. 7). After IsoNet missing-wedge correction, interpretability has improved as judged by readily recognizable structures such as vesicles, microtubules, and scattered complexes in all three orthogonal views of the tomograms (Fig. 7).

## Discussion

Here we have developed a deep learning-based package, IsoNet, to overcome the limitation of missing-wedge problem and low SNR plaguing all current cryoET methods. To demonstrate its robustness, we have applied IsoNet to process three representative types of cryoET data−isolated ribosomes and viruses, cellular organelle axoneme with PFR, and neuronal synapse−representing three levels of complexity. IsoNet significantly improved structural interpretability in all these cases, allowing us to resolve novel structures of lattice defects in immature HIV capsid, dynein subunits, and scissors-stack-network architecture of the paraflagellar rod in eukaryotic flagella, and heptagon-containing clathrin cage inside a neuronal synapse. In the resulting tomograms, the in situ protein features appear isotropic and have high quality that sometimes matches that obtained through subtomogram averaging. For amorphous structures in the tomograms, such as membranes, IsoNet allows the network to learn the feature representation from many other similar structures in the tomogram and recover the missing information. Thus, IsoNet expands the utility of cryoET by overcoming its inherent missing-wedge problem, enabling 3D visualization of structures that are either complex as those in cells (Figs. 5, 6) or are rare as those in tomograms of patient tissues[61]. Notably, IsoNet outperforms program packages ICON[18] and MBIR[19] designed to deal with missing-wedge problems (Supplementary Fig. 6).

Philosophically speaking, no information can emerge from a vacuum/nothing. Where does IsoNet recover the missing information from? The questions touch upon the fundamentals of deep learning and can be thought of as relating to the non-locality of information in space. By learning from information scattered around in original tomograms with recurring shapes of structures, IsoNet sophistically eliminates distorted or missing information. The great advantage of the IsoNet approach is that similar features across different dimensions can be automatically discovered and "averaged" without human intervention. Such features could be related in translation and rotation manners in the three Cartesian dimensions, such as crystalline PFR subunits and axonemal microtubules (Fig. 4); they could also be related through symmetries, such as those pentagons and hexagons of clathrin cages (Fig. 6); they could be related biologically, such as the facts that proteins are made up of only 20 amino acids and nucleic acid of four bases, and both are geometrically constrained as a linear molecule; ultimately, they could also be related to compositionality of the natural signals, such as the horizontally orientated carbon film can be decomposed into common features, including dots, lines, and planes (Fig. 2). IsoNet learns their relationships in the same tomogram or across multiple tomograms and reconstructs these features automatically. In essence, IsoNet and subtomogram averaging compensate for the missing-wedge problem through the same principle.

Although overfitting should always be considered when dealing with noisy data, such as cryoET data. IsoNet, in principle, won't be confused between similar molecules to cause overfitting. This is because missing information recovery is performed across multiple scales in the IsoNet neural network, which begins with detecting features in a very local area (e. g., $3 \times 3 \times 3$ voxels) towards a very large (e.g., $64 \times 64 \times 64$ voxels). Even the slightest differences among similar molecules would dominate at a small scale. And in that scale, the IsoNet algorithm will penalize artificially adding small components to a molecule.

Regardless of the details of information recovery, the substantial improvement in map interpretability afforded by IsoNet now allows visualization of structures for functional interpretation without the need for tedious and time-consuming subtomogram averaging, which typically involves a priori feature identification and manual particle picking. Visualizing such structures in cellular tomograms by IsoNet would also improve localization and subsequent subtomogram averaging of hundreds of thousand copies of like-structures, leading to in situ atomic resolution structures of cellular complexes in their native cellular environment.

## Methods

### Software implementation

We implemented IsoNet in Python using Linux as the native operating system. Typical hardware setup includes one node with four Nvidia GeForce 1080Ti GPU cards of 11 gigabytes of memory, which is common in a cryoET research laboratory. The package can be run from the command line and relies on Keras, which acts as an interface for Tensorflow[38], and the package can be downloaded from Github (https://github.com/IsoNet-cryoET/IsoNet). A detailed document is provided, accompanied by the IsoNet software. The tutorial dataset and video can be found in the GitHub repository.

This package is standard-alone and does not rely on other software such as IMOD[62], while some common Python modules must be installed prior to running IsoNet. Such Python modules are easy to install with the "pip" command. For example, the IsoNet uses the Python module "mrcfile" to read and write tomogram or subtomogram, and "numpy" for image processing such as rotation and Fourier transform. The U-net neural network is built by stacking multiple layers (Supplementary Fig. 3) provided in "tensorflow.keras.layers". For example, three "Conv3D" layers are stacked together in each depth of the encoding path of the U-net.

The package can be launched through a single command entry, either "*isonet.py*" or "*isonet.py gui*", for Linux command line operations or a graphical user interface (GUI) (Fig. 1b), respectively. Users can then

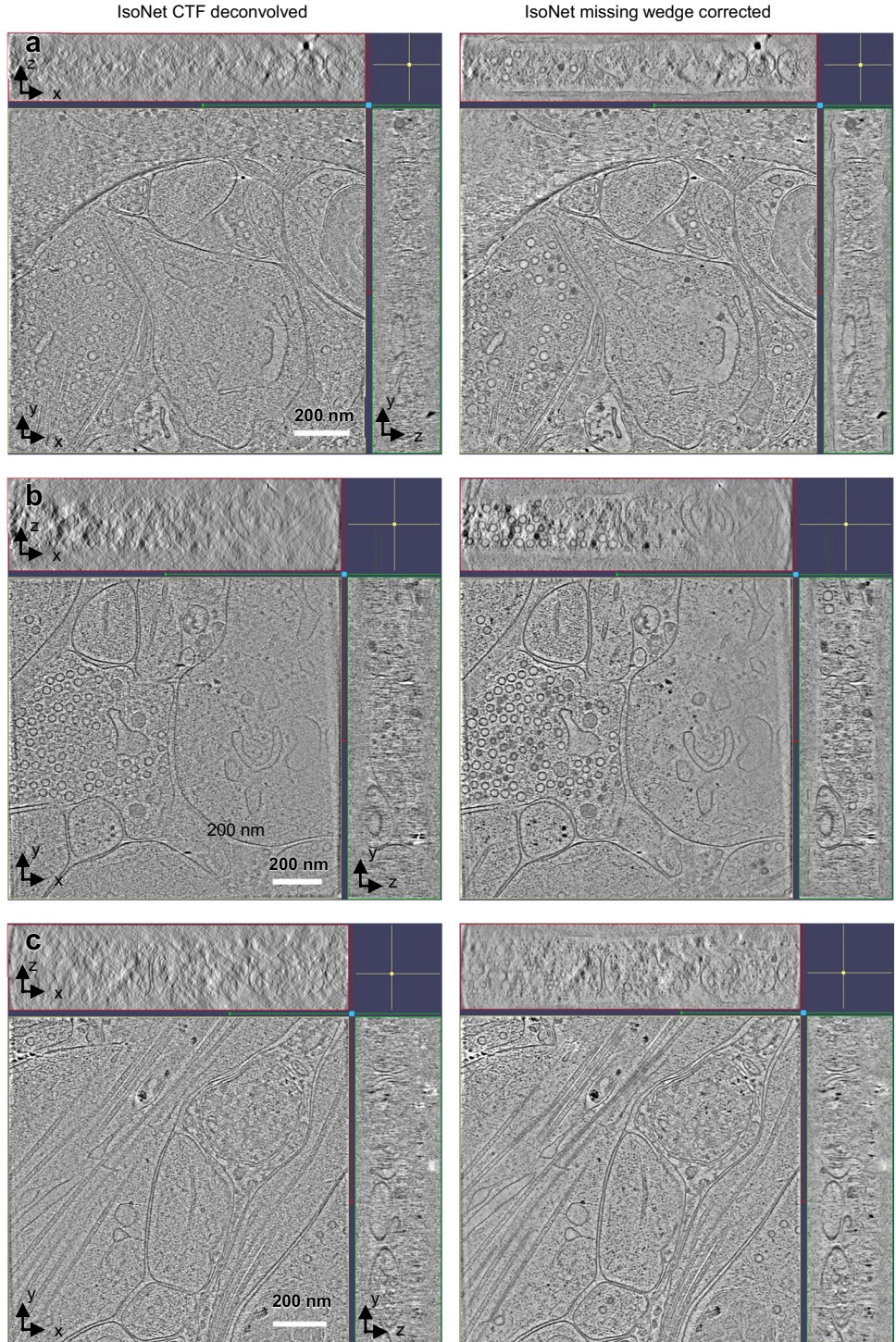

**Fig. 7 | The performance of IsoNet for low SNR cellular tomograms. a–c** Orthogonal slice views of CTF deconvolved (left), and missing-wedge corrected (right) synaptic tomograms.

access all the processing steps of the IsoNet procedure. The IsoNet procedure contains five steps, including three major steps: *Extract, Refine,* and *Predict* and two accessory steps: *CTF deconvolve* and *mask generate.* Each of these steps corresponds to one command of IsoNet in the Linux terminal and will be described in the following sections.

**Dataset preparation**

To use IsoNet, users should prepare a folder containing all tomograms. Binning the tomograms to a pixel size of about 10 Å is recommended. Typically, a folder containing 1 to 5 tomograms is used as input.

These input tomograms can either be reconstructed by SIRT or WBP algorithm. The tilt axis is the Y-axis, and recommended tilt range is from −60° to 60°, while other tilt ranges might also work. The tilt series can be collected with any tilt scheme, continuous, bidirectional, or dose-symmetric.

IsoNet uses a Self-defining Text Archive and Retrieval (STAR) file format to store information of tomograms and program parameters used during Isonet processing. Thus, it inter-operates seamlessly with such leading cryoEM software packages as Relion[40]. Tomogram STAR file, by default named as *tomograms.star*, is required to run IsoNet, and

this file can be prepared with IsoNet GUI, with a text editor, or with the following command:

$$\$ isonet.py \ prepare\_star \ [tomogram\_folder]$$

## Deconvolve CTF

For the tilt series imaged without a Volta phase plate (VPP), the sinusoidal CTF suppresses or even inverts information at certain frequencies. To enhance the contrast of the tomograms and promote information retrieval, CTF deconvolution, similar to what is implemented in Warp software, is applied to tomograms in this step.

IsoNet uses a Wiener-like filter[28] for CTF deconvolution, with a spectral signal-to-noise ratio (SSNR) set empirically:

$$SSNR = e^{-f \times 100F} \times 10^{S} \times H_{f} \qquad (1)$$

Where f denotes the spatial frequency, H is a high-pass filter, F is the custom fall-off parameter, and S denotes the custom strength parameter. Because the SSNR of the Wiener-like filter is determined empirically, users can tune the *snrfalloff* or deconvolve strength parameters to enhance the contrast of the tomograms. This step can be performed by the "CTF deconvolve" function in GUI or with the following command:

$$\$ isonet.py \ deconv \ [tomogram\_star] \ \text{--}snrfalloff \ 1.0$$
$$\text{--}deconvstrength \ 1.0$$

## Generate mask

Subtomograms for training would be better to contain rich information than empty areas with only ice, air, or carbon. In this optional mask generation step, IsoNet uses statistical methods to detect empty regions from which subtomograms will not be extracted. Two different masks can be applied: pixel intensity masks that exclude areas with low cryoET density and standard deviation masks that exclude areas with low standard deviation.

The pixel intensity mask will first suppress the noise with a Gaussian filter and then apply a sliding window maximum filter to the contrast-inverted tomogram (i.e., write density and black background). The areas with relatively smaller density values in the filtered tomogram will be deemed as empty space. The parameter "*density_percentage*" defines the percentage of the area kept in the tomogram by the density mask.

The standard deviation mask is achieved by calculating the standard deviation of voxels in a small cubic volume centered at each evaluating voxel. The voxels having a relatively lower standard deviation will be excluded. The parameter "*std_percentage*" defines the percentage of voxels kept by the standard deviation mask.

IsoNet uses the intersection of these two masks to exclude empty areas. The parameters for mask generation can be tuned to cover the region of interest but exclude empty areas. In addition to these two types of masks, IsoNet allows excluding top and bottom parts of tomograms which usually are empty areas, by the "*z crop*" parameter.

Usually, the default parameters will provide a good mask to exclude empty areas; users can also tune the parameters using GUI by the "Generate mask" function or the following command, for example:

$$\$ isonet.py \ make\_mask \ [tomogram\_star] \ \text{--}density\_percentage \ 50$$

## Extract subtomograms

In each tomogram, a specified number of seeds are randomly generated within the whole tomogram or the region of interest defined by the mask. Then, cubic volumes centered at the generated seeds are boxed out and saved as subtomograms. The extracted subtomograms should be large enough to cover typical features in tomograms, such as a patch of membrane or vesicle. However, due to the GPU memory limitation, this size cannot be arbitrarily large. We usually extract 300 subtomograms of $96^{3}$ voxels in total. After extraction, the contrast of those subtomograms is inverted. Then, tomograms are normalized by percentile to ensure that 90% of the voxel values fit into the range between zero and one. The subtomograms can be randomly split into two halves and used for performing missing-wedge correction independently to eliminate overfitting and calculate gold-standard FSC.

The information of extracted subtomograms is stored in another STAR file, default named *subtomo.star*. Subtomogram extraction can be performed by either IsoNet GUI or the following command:

$$\$ isonet.py \ extract \ [tomogram\_star]$$

## Refine

This process iteratively trains neural networks to fill the missing-wedge information using the same tomograms whose missing-wedge artifacts were added to other directions. The denoising module can also be enabled in this step, making the network capable of reducing noise and recovering the missing-wedge. After refinement, the resulting subtomograms and neural network model in each iteration are saved. The network models with a suffix of ".h5" can be used for the *Predict* step.

Four steps, including *training dataset generation*, *adding noise*, *network training*, and *subtomograms prediction*, will be performed during each iteration. These steps will be described in the following sections. The missing-wedge restored subtomograms by *subtomograms prediction* in every iteration will be used for *training dataset generation* in the next iteration. Usually, 10–15 iterations in the *Refine* step are sufficient to obtain a well-trained network for the missing-wedge correction, whereas more iterations can be performed for refinement with denoising.

The *Refine* step can be performed from the GUI or with the following command, for example:

$$\$ isonet.py \ refine \ [tomogram\_star] \ \text{--}iterations \ 30 \ \text{--}gpuID \ '0,1,2,3'$$

Users can also continue training from the previous iterations using "*continue from*" or from previously trained models using "*pretrained_model*" parameter.

### Refine step 1: training dataset generation

IsoNet rotates the extracted subtomograms to different orientations to generate paired datasets for neural network training. Twenty rotated copies can be obtained for each extracted subtomogram as follows (Supplementary Fig. 2). First, each subtomogram is a cube with six faces. Each face can be rotated with an out-of-plane angle to face toward the positive direction of the Z-axis. Second, each out-of-plane rotation can be followed by four in-plane rotations, making 24 possible rotations. However, four of the 24 rotations result in subtomograms with the same missing-wedge direction as the original subtomograms. Thus, these four rotations are excluded, resulting in 20 orientations for each subtomograms. This rotation process enlarges the original dataset by 20 times for training, making it possible to achieve a good performance of missing-wedge correction even with a small dataset, e.g., a single tomogram.

After the rotation, the IsoNet program then applies the missing-wedge filter to the rotated subtomograms. The missing-wedge filter volume has the same size as that of the subtomograms. In the missing-wedge filter volume, the voxel value is zero inside the missing-wedge region and one in the rest of the volume. Then, the Fourier transforms of the rotated subtomograms are multiplied by the missing-wedge

filter volume and then transformed back to the real space, generating missing-wedge filtered subtomograms.

To avoid incomplete information along the edge of the subtomograms when applying the missing-wedge filter, both rotated subtomograms and missing-wedge filtered rotated subtomograms are trimmed into smaller volumes (often $64^3$ voxels), generating "target" and "input" for the network training, respectively. These generated data pairs are used to train a neural network that maps the "input" to "target".

## Refine step 2: adding noise

This optional step allows performing missing-wedge correction and denoising simultaneously using IsoNet. IsoNet uses a noisier-input strategy[42,43] that learns to map "input" with additional noise to the "target".

IsoNet simulates the noise pattern in reconstructed tomograms with the assumption that the noise in tomograms is an additive Gaussian noise in every acquired projection and independent among all images acquired in a tilt series. During the adding noise step, a set of 3D noise volumes are constructed by back-projecting a series of 2D Gaussian noise images to reflect the effect of the backprojection algorithm on noise formation.

In IsoNet, we provided three noise models for various tomogram reconstruction algorithms: simple backprojection, backprojection with ramp filter, and backprojection with hamming filter. These noise models correspond to three tomograms reconstruction methods in IMOD: SIRT, weighted backprojection, and reconstruction with a hamming-like filter, respectively. Users can choose different noise reconstruction methods for their tomograms accordingly.

The denoise level is defined as the ratio of the standard deviation between the added noise and the subtomograms. The noise volumes are scaled to match the denoise level before adding to the "input" subtomograms. Thus, the lower denoise value means less noise is added to individual subtomograms.

Because the added noise may further corrupt the 3D subtomograms, making the network hard to train, it is recommended to start the first several iterations of refinement without denoising. After the refinement results are nearly converged, the noise volume can then be added to the "input" subtomograms in the following iterations. A typical routine is to train ten iterations without denoising and then increase the denoise level by 0.05 for every five iterations. This step-wised noise addition can be performed automatically in the *Refine* step of the IsoNet software.

## Refine step 3: network training

The neural network used in IsoNet is based on U-net, which is well recognized in biomedical image restoration and segmentation[41]. The main building blocks of the U-net are 3D convolution layers with non-linear activation functions called Rectified Linear Units (Relu), which are applied per voxel. Those convolution layers have kernel sizes of $3 \times 3 \times 3$. Three 3D convolution layers are stacked together to form a convolution block in our network, which can extract complicated features.

By stacking the convolutional blocks, the U-net is built based on encoder-decoder architecture (Supplementary Fig. 3). The encoder path is a set of convolution blocks and strided convolution layers that compress 3D volumes. Strided convolution layers reduce the spatial size of the input of this layer by $2 \times 2 \times 2$, allowing the network to learn more abstract information. A convolution block followed by a strided convolution layer makes one encoder block in the contracting path. A total of three encoder blocks form the entire encoding path. The number of convolution kernels for each convolution layer doubles after each encoder block. After the encoder path, the 3D volumes are processed with a convolution block and enter the decoder path of the network. The decoder path is symmetrical to the encoder but uses transpose convolution layers, opposite to strided convolution layers, to enlarge the dimension of features.

Although the downsampling of the 3D volumes captures the essence of the features, high-resolution information is lost by stride convolution operations. In particular, the skip-connections that concatenate the feature layers of the same dimension in two paths are implemented to preserve the high-resolution information. A dropout strategy that randomly sets 50% of neurons' activation to 0 in the convolution layers is used to prevent overfitting during the training.

This network uses the mean absolute error between the output of the network and the target subtomograms as a loss function. The loss function is minimized by employing Adam optimizer[63] with an initial learning rate of 0.0004. The neural network training in each iteration is performed on GPU and consists of ten epochs. Each epoch will traverse through the paired dataset. The data pairs are grouped into batches (which generally have a size of 8 or 16) to feed into each epoch. After the training, the trained neural networks are saved for the next iteration of the *Refine* step.

## Refine step 4: subtomogram prediction

After each iteration of refinement, the network is applied to the original subtomograms, generating predicted subtomograms. Then IsoNet generates an inverse missing-wedge filter volume with ones inside the missing-wedge region but zeros in the rest of the 3D volume. The predicted subtomograms are then transformed to Fourier space and multiplied with the inverse missing-wedge filter volume to extract the added information inside the missing-wedge region. Then, these filtered volumes are added to the original subtomograms, generating the missing-wedge restored subtomograms for subsequent refinement iterations.

## Predict

After the *Refine* step, the trained network is saved in a model file. It will be used to correct the missing-wedge for the original tomogram or other similar tomograms. For most tomograms, the full-size 3D images can hardly fit into the memory of a regular GPU. Thus, the IsoNet program splits the entire tomogram into smaller 3D chunks and applies the network to them separately. Then output 3D chunks are montaged to produce the final output. To avoid the line artifact between adjacent chunks caused by the loss of information on the edges of subtomograms. We implemented a seamless reconstruction method called overlap-tile strategy[41], which predicts the overlapping chunks to avoid the edge effect. The "crop_size" parameter defines the size of the cubic chunks. This *Predict* step can be performed with IsoNet GUI or with the following command, for example:

$isonet.py\ predict\ [tomogram\_star][output\_folder]\ \text{--}model$
$[network\_model]$

## Benchmarking with simulated data

We performed IsoNet reconstruction on simulated subtomograms using publicly available atomic models: apoferritin model[44] (PDB: 6Z6U) and ribosome model[45] (PDB: 5T2C). For both tests, density maps were simulated from the atomic models using "*molmap*" function in ChimeraX[46] and filtered to 8 Å resolution (Fig. 1d, e). Those simulated maps with 2.67 Å/pix pixel size were then rotated in ten random directions and imposed with missing-wedge filter in Fourier space, resulting in simulated subtomograms with missing-wedge artifacts (leftmost columns in both Fig. 1d, e).

For the simulated Apoferritin maps, we created a subtomogram STAR file with the "*isonet.py prepare_subtomo_star*" command. With this subtomogram STAR file as input, we performed IsoNet *Refine* step for ten iterations without denoising. For benchmarking with the simulated ribosome maps, we extracted eight smaller subtomograms

(96³ voxels) from each ribosome map due to the larger dimension of the ribosome map. The subtomogram STAR file generated in the extract step was used for the subsequent *Refine* step. After ten iterations, a trained network was obtained and was then used to produce missing-wedge corrected maps of ribosome using "isonet.py predict" command.

## Processing tomograms of ribosomes

We downloaded the dataset of ribosome tomograms from EMPIAR-10045[48]. The total seven tomograms were binned six times to reach a pixel size of 13.66 Å/pixel. Those tomograms are then CTF deconvolved with *snrfalloff* parameter 0.75 in IsoNet. For each tomogram, a mask was generated using default parameters in IsoNet to exclude empty regions, using the "isonet.py make_mask" command. From those seven tomograms, 490 subtomograms with a box size of 80³ were extracted.

Then, we executed the "IsoNet.py refine" command to train the network for 30 iterations. We used the default denoise parameters in IsoNet. The final denoise level reaches 0.2. Then, we performed the missing-wedge correction for those seven CTF deconvolved tomograms with the "Isonet.py predict" command, using the trained network in the *Refine* step.

## Processing tomograms of the HIV virus

For pleomorphic viruses, we downloaded an HIV dataset from the public repository EMPIAR-10164[11]. Three tilt series, TS_01, TS_43, and TS_45, were used for testing. The movie stacks were drift-corrected with MotionCorr[64] and reconstructed with IMOD[21] using the WBP algorithm. The defocus value of each image was determined by CTFFIND4[65]. Eight-time binned tomograms with 10.8 Å pixel size were used for further processing. For the CTF deconvolution of the tomograms, the SSNR fall-off and the deconvolve strength parameters were set to 0.7 and 1, respectively. Then, we created one mask for each tomogram using the "isonet.py make_mask" command. A total of 300 subtomograms with 96³ voxels were randomly extracted from the three tomograms and then split into random halves. For each half of the subtomograms, we performed *Refine* step for 35 iterations independently, generating two trained neural networks. In the *Predict* step of IsoNet, tomogram TS_01 was used to generate two missing-wedge corrected tomograms by the two independently trained networks. These two tomograms were then averaged to create a final map.

These two missing-wedge corrected tomograms enabled calculating gold-standard FSC. Instead of calculating a global FSC, we performed a 3D FSC calculation for all the directions[39] to measure the resolution anisotropy of the missing-wedge corrected tomogram. Because the 3D FSC calculation works for cubic volumes while the size of the tomogram is non-cubical, we cropped the generated tomograms into cubic subtomograms for the 3D FSC calculation. As for the HIV dataset, the 3D FSC was calculated for four 200³ volumes cropped from both missing-wedge corrected HIV tomograms. The resulting four 3D FSC were then averaged to produce the final 3D FSC, the orthogonal sections of which are shown in Fig. 3b.

## Processing tomograms of the Eukaryotic flagella

For cellular organelles, we chose the demembraned flagella of *Trypanosoma Brucei*. The datasets described here were obtained in our previous studies[51,66]. Tilt series were recorded with SerialEM[67] by tilting the specimen stage from −60° to +60° with 2° increments. The cumulative electron dosage was limited to 100 to 110 e⁻/Å² per tilt series. The movie stacks were drift-corrected with MotionCorr[64] and reconstructed with IMOD using the SIRT algorithm. The tomograms were binned by four, resulting in a pixel size of 10.21 Å/pixel.

Three tomograms were chosen for missing-wedge correction. These tilt series were acquired with VPP, so we did not perform the CTF deconvolution. We generated one mask for each tomogram using the "isonet.py make_mask" command. Then, we extracted a total of 360 cubic subtomograms with 128³ voxels from three tomograms. Using these subtomograms, we trained a network by running the *Refine* step for 35 iterations with default denoise levels, which were automatically changed from 0 to 0.2. The trained network produced in the *Refine* step was then used to run the *Predict* step of IsoNet to obtain a final missing-wedge corrected tomogram, which is shown in Fig. 4.

## Processing tomograms of hippocampal neurons

Tomograms of hippocampal neurons were obtained in our previous study[17]. The tomograms used in this study were collected on a Titan Krios microscope equipped with K2 summit in counting mode. The energy filter (Gatan image filter) slit was set at 20 eV. The Titan Krios was operated at an acceleration voltage of 300 KV. Tilt series were acquired using SerialEM[67] with tilt scheme: from +48° to −60° and from +50° to +66° at an interval of 2°. The total accumulated dose was -150 e-/Å². The pixel size of the tomograms is 4.35 Å/pixel. Each recorded movie stack was drift-corrected and averaged to produce a corresponding micrograph using MotionCorr[64]. The tilt series were aligned using IMOD[21]. One tilt series of the tomogram shown in Fig. 5 was imaged with VPP, while the other shown in Fig. 6 was acquired without VPP. When VPP was used, the defocus value was maintained at −1 μm; otherwise, it was kept at −4 μm.

For the tilt series recorded with VPP, the aligned tilt series were reconstructed using NovaCTF[68], generating a tomogram reconstructed with WBP. The tomogram was binned by four, and 300 subtomograms (96³ voxels) were extracted from that tomogram. Those subtomograms were then used for 35 iterations of refinement. The trained network produced in *Refine* step was used for missing-wedge correction for the entire tomogram (Fig. 5).

The one tomogram containing the clathrin cage is recorded without VPP. The defocus value of each image was determined by CTFFIND4[65], and the CTF phase flipped tomogram was obtained by NovaCTF[68]. This tomogram (Fig. 6) was reconstructed with a SIRT-like filter, with CTF phase flipping performed on the individual tilt images. The tomogram was binned by four for missing-wedge correction with IsoNet. Then, 200 subtomograms (96³ voxels) were extracted from the tomogram in the extract step of IsoNet. A trained network was obtained with the *Refine* step of IsoNet for 35 iterations. The trained network was then used for the *Predict* step of IsoNet, producing a missing-wedge corrected tomogram (Fig. 6).

For the missing-wedge correction of the neuron dataset in Fig. 7, we chose three tomograms whose tilt series covers −60° to 60° range and without the x-axis tilt. The tomograms were reconstructed with the SIRT algorithm in IMOD for five iterations. The tomograms are then four times binned, resulting in a pixel size of 17.4 Å/pixel, using "*binvol*" command in IMOD. To further increase the contrast of the tomogram, we performed CTF deconvolution using IsoNet. Then, for each tomogram, we extracted 100 subtomograms with 96³ voxels. A trained network was obtained with the *Refine* step of IsoNet for 30 iterations with default denoise levels, which were automatically changed from 0 to 0.2. The generated neuron network can be applied to other similar neuron tomograms (Fig. 7).

## 3D visualization

IMOD[62] was used to visualize the 2D tomographic slices. UCSF ChimeraX[46] was used to visualize the resulting IsoNet-generated tomograms in their three dimensions. Segmentation of density maps and surface rendering were performed by the volume tracer and color zone in UCSF ChimeraX. The atomic models of ribosomes[45] were fitted to the tomograms using the "fit in map" tool in UCSF ChimeraX.

## Reporting summary

Further information on research design is available in the Nature Research Reporting Summary linked to this article.

## Data availability

Dataset[69] used in this study, including three tomograms of HIV particles, one IsoNet corrected HIV tomogram, one original tomogram of a neuronal synapse with its IsoNet reconstruction, and ten simulated apoferritin subtomograms is deposited into the publicly available repository Figshare (https://figshare.com/articles/dataset/Dataset_to_reproduce_results_in_IsoNet_paper/20560443).

## Code availability

The software has been deposited to the GitHub page at https://github.com/IsoNet-cryoET/IsoNet.

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

## Acknowledgements

We thank Jason Hu for testing the program with the ribosome dataset, Chong-Li Tian, Yi-Nan Xia, and Zhen-Hang Lu for testing the program with the neuron dataset, and Jiayan Zhang, Simon Imhof, Ivo Atanasov, Wong Hui, and Kent Hill for the Eukaryotic flagellum data. This work was supported in part by grants from the National Key R&D Program of China (2017YFA0505300 to G.-Q.B.), the National Natural Science Foundation of China (31630030 to G.-Q.B., 31761163006 to G.-Q.B., and 31621002 to G.-Q.B.), the Strategic Priority Research Program of the Chinese Academy of Sciences (XDB32030200 G.-Q.B.). Research in the Zhou group is supported in part by the US National Institutes of Health (GM071940 to Z.H.Z.). We acknowledge the use of resources at the Center for Integrative Imaging of Hefei National Research Center for Physical Sciences at the Microscale, and those at the Electron Imaging Center for Nanomachines of UCLA supported by US NIH (S10RR23057 to Z.H.Z. and S10OD018111 to Z.H.Z.) and US NSF (DMR-1548924 to Z.H.Z. and DBI-133813 to Z.H.Z.).

## Author contributions

Y.-T.L. conceptualized the method, wrote code, processed data, made illustrations and documentation, and wrote the paper; H.Z. wrote code, processed the HIV and neuronal synapse data, made documentation and assisted in illustrations, and wrote the paper; H.W. participated in coding, processed the axoneme with PFR data, made documentation, and assisted in illustrations; C.-L.T. assisted in testing the method with the neuronal synapse data and in making illustration based on the test results; G.-Q. B. and Z.H.Z. oversaw the project, interpreted the results, and wrote the paper. All the authors edited and approved the manuscript.

## Competing interests

The authors declare no competing interests.
