## [Peer Review File · Nature Communications]

Isotropic Reconstruction for Electron Tomography with Deep LearningREVIEWER COMMENTS

Reviewer #1 (Remarks to the Author):

This manuscript presents a deep learning based model, IsoNet, for compensating the missing wedge effect in cryo-ET data. As a novel deep learning based method for this task, the method clearly demonstrated better performance than previous traditional methods such as ICON and MBIR. The authors did a good job of explaining the underlying methodology with informative conceptual figures. To evaluate IsoNet, the authors applied it to one benchmarking simulated dataset and three real datasets. The results showed that IsoNet is able to fill the missing information that is biologically meaningful, which would help structural biologists to better interpret the rich information in cryo-electron tomograms. I have some concerns which would help the demonstration and presentation of IsoNet if addressed.

Major concerns:

1. One special design is the iterative refine step. However, would error accumulate in the iterative process? Please have a discussion on this.
2. The benchmarking is done on a simulated dataset. I suggest using/adding a real single-particle dataset, for example, EMPIAR 10045, the benchmark dataset for Relion, to do the benchmarking. The authors could compare the missing wedge effect at different particle orientations, their corrected structures, and the structures recovered from averaging.
3. The philosophical discussion in the Discussion section is very meaningful. However, this leads to a question of whether IsoNet can do a good job with highly diverse structures. I noticed all three real datasets contain relatively simple structures with many repetitive structural patterns. But a lot of tomograms contain very crowded and diverse subcellular structures. Therefore, it would be helpful if the authors can find a subcellular dataset with diverse structures in a crowded cytoplasm environment to show IsoNet's performance.
4. About the application of IsoNet, facilitating particle picking is definitely an important one. For example, using Laplacian of Gaussian as mentioned by the authors. I think the authors can do a simple experiment on a single particle dataset, for example, EMPIAR 10045, by comparing Laplacian of Gaussian particle picking on tomograms with and without IsoNet correction. The results can be evaluated by whether subtomogram averaging improves from the picked particles.
5. Missing discussion of time cost: training time and prediction on tomograms. Also, does each iteration contain one epoch training?

Minor concerns:

1. Some grammar errors: for example Ln 89 'In the cryo-EM field', Ln 91 'the missing wedge information'
2. No previous deep learning works for cryo-ET tasks are discussed.
3. Ln 349-354, these are trivial details that can be moved to the Results or Methods section.
4. Ln 775, please have a citation for overlap-tile strategy.
5. Figure 1 caption missing e.

Reviewer #2 (Remarks to the Author):

In this paper, Liu et al present an AI-based approach for reducing anisotropy of cryo-tomographic reconstructions. The isonet program aims to fill the 'missing wedge' by extracting subtomograms at random points in the tomograms, and comparing them in a number of rotations to detect the common information, thereby deriving the missing information from mutually similar objects in different orientations.

The software can be run on the command line or through a gui, and includes steps of deconvolution (similar to warp, producing high SNR tomograms), masking (identifying regions of the tomograms likely to contain relevant density), subtomogram extraction, training, and predicting.

The results presented in the paper are impressive, suggesting this approach is more powerful than others previously proposed.

We've actually tried isonet on our own tomograms of reconstituted COPII vesicles and were very impressed with the results, with membranes becoming visible along the z direction, and many features becoming better resolved and directly interpretable. In our isonet-treated tomograms we have been able to visualise individual COPII inner coat complexes on the surface of vesicles which we were unable to do before, no matter what filter we tried. Moreover, with isonet we were able to segment fully closed membranes semi-automatically by simply setting a threshold in chimera, which is allowing us to streamline the particle-picking process significantly.

However, we have tried the software on cellular tomograms, which are much more heterogeneous, and could not see a significant improvement. The same was reported by other colleagues working with in situ data. This differs from the results presented in the paper where clearly an effect was seen for cellular data.

As far as I understand, isonet works by identifying groups of subtomograms that contain similar features, such as membranes, so is it possible that in an environment that is too crowded with heterogeneous features the performance is lower?

I am happy to recommend the paper for publication, but I would like to see a detailed description of the protocol and parameters used to optimise the in-cell data, with a discussion of potential difficulties related to heterogeneous samples. Could the authors suggest optimum settings for 'clean' purified samples and heterogenous in situ data (or provide details of the settings used for each individual dataset)?

Other comments:

1. We played quite a bit with the masking step but did not succeed in masking out the carbon. We had to trim the volume to include the central portion only, use the trimmed tomogram for the training set, and then predict on the full tomograms. It would be useful if in addition to crop_z, also a crop_x and crop_Y options were added to mask out peripheral areas, so these manual cropping steps can be avoided.

2. In the introduction, other methods of recovering the missing wedge are dismissed: "However, such assumptions have limited information content (or "entropy") and may not always hold true, given the complexity of biological systems.". The authors should explain with more detail why these methods have limited information content, and how IsoNet does not suffer from the same shortcomings, either in the introduction or discussion.

3. It should be stated in the abstract and at the end of the introduction that IsoNet works for tomograms at a pixel size of $\sim 10 \text{ \AA}$ and above to avoid confusion that it can be used on high-resolution tomograms.

4. Can the authors discuss whether there could be overfitting/confusion if similar molecules are present in the tomogram? Could information from different but similar molecules be 'mixed' to fill in missing wedges of similar ones?

5. Denoising: will the simulated backprojected noise be a good model independently of the reconstruction algorithm (i.e. sirt versus backprojection)?

Minor Points:

1. Line 636: contrast inverted with respect to what? i.e. are users expected to start with black density and will the starting contrast affect the result?

2. When describing the refine 1 step details, it needs to be specified that the MW is applied along the zy axis of the unrotated tomograms. As it reads currently, I was under the impression the MW is applied along the same direction as the 'true' MW, i.e. rotated together with the tomogram, which doesn't make sense. Fig 1c, top panel, clarified that for me but it needs to be written clearly too.

3. "Its application to high-resolution cellular tomograms should also help identify differently oriented complexes of the same kind for near-atomic resolution sub-tomogram averaging." – Remove 'near-atomic' as this claim is substantial and requires proof.

4. The paper contains grammatical errors and imprecisions, please re-read the paper and correct them. Some are listed below, but there were too many to comprehensively correct all of them. Also, references are missing, please see below for details.

- "Examples abound, ranging from pleomorphic viruses, to cellular organelles, to large-scale cellular structures like synapses between neurons." – Cite a range of papers here.

- "Many viruses, notably those involved in devastating pandemics such as SARS-CoV-2, influenza viruses, and human immunodeficiency viruses (HIV), are pleomorphic in the organizations of their proteins and genomes." – Again, cite references.

- "This technique requires collecting a series images of the sample at different tilt angles, called "tilt series"." – Should be: 'called a "tilt series"'.
- "low signal-to-noise ratio (SNR) for the cryo tomogram." – Should be 'in the cryo tomogram'.

- The paragraph beginning: "To reveal such molecular sociology across viruses or inside cells, cryogenic electron tomography (cryoET) has become the tool of choice." Should contain more references, see cryo-ET reviews (e.g. Saibil & Orlova, Wan & Briggs).

- "the tilt range for cryoET is usually restricted to about $\pm 70^\circ$." - 60° is the more commonly used tilt range.

- "to constraint the structural features in reconstructed tomograms." – Should be "to constrain".

- "In the field of computer vision, convolutional neural network (CNN) has been applied to various tasks, such as object recognition, image segmentation, and classification, often achieving high performance" – Again, cite references.

- "Here, we have developed a CNN-based software system, called IsoNet, for isotropic reconstruction of electron tomogram." – Should be 'reconstruction of tomograms'.

- "The resolution at Z-axis reaches about 30\AA resolution as measured by the gold-standard Fourier shell correlation (FSC) criterion." – Replace 'at the Z-axis' with 'within the missing-wedge'.

- "thousands of sub-tomogram particles for future near-atomic resolution cryoET studies." – Replace 'near-atomic' with 'high resolution'.

- "Thus, it is possible to recover the missing information by merging information from similar features present in the same tomograms but at different orientations relative to each other." – Remove 'thus'.

- "Among the 5 steps, Refine and Predict relies on graphical processing unit (GPU) that provides

superior processing power.” - Should be: “Among the 5 steps, Refine and Predict requires GPU acceleration”.

- “particularly on Z-axis.” – Should be ‘particularly in the Z-axis’.

- “To further improve miss-wedge correction” – Should be: ‘missing-wedge’.

- “ribosome as the second test due to its asymmetric shape and primarily nucleic acid content.”. The ribosome is not formed primarily of nucleic acids, remove this part.

- “Importantly, our isotropic 3D reconstruction shows that the quality of the structure is similar across all directions, allowing biological structures to be interpreted adequately (Fig. 2c and Supplementary Video 1).” – This is true for low resolution features only and this should be clarified.

- “Next, we tested the performance of IsoNet on resolving structures within cellular organelles by processing tomograms of flagella of Trypanosoma. Brucei using IsoNet.” – Please cite the paper where these tomograms were produced here and remove the ‘.’ After Trypanosoma.

- Correct all instances of ‘Weiner’ to ‘Wiener’

Response to Reviewers' Comments

Summary of responses: We thank the reviewers for their supportive comments and constructive suggestions. As can be found in our itemized responses below, we have taken several months to carefully consider the reviewers' comments and thoroughly address their concerns by performing the suggested tests and improving the user manual. Four major efforts have been undertaken: 1. Processed ribosome dataset EMPIAR-10045 with IsoNet and compared the IsoNet densities with atomic model (new figure 2); 2. Showed examples of heterogeneous cellular tomograms reconstructed with IsoNet (new figure 7); 3. Add a paragraph for optimized parameters for cellular tomography; 4. Wrote more details in the IsoNet tutorial and the method section of the revised paper.

To facilitate your navigation of this document, the reviewers' original comments are pasted in **black**, and our responses are in **blue**.

REVIEWER COMMENTS

Reviewer #1 (Remarks to the Author):

This manuscript presents a deep learning based model, IsoNet, for compensating the missing wedge effect in cryo-ET data. As a novel deep learning based method for this task, the method clearly demonstrated better performance than previous traditional methods such as ICON and MBIR. The authors did a good job of explaining the underlying methodology with informative conceptual figures. To evaluate IsoNet, the authors applied it to one benchmarking simulated dataset and three real datasets. The results showed that IsoNet is able to fill the missing information that is biologically meaningful, which would help structural biologists to better interpret the rich information in cryo-electron tomograms. I have some concerns which would help the demonstration and presentation of IsoNet if addressed.

Response: Thank you for your comments and suggestions on our manuscripts.

Major concerns:

1. One special design is the iterative refine step. However, would error accumulate in the iterative process? Please have a discussion on this.

Response: The answer is NO. To prevent error accumulation, the predicted subtomograms in previous iterations are not directly used for the next iteration. Instead, the subtomograms are prepared by adding the missing wedge information of corrected subtomograms to the original subtomograms. This means any information from the original subtomograms will not be modified throughout the iterative process, ensuring the fidelity of the IsoNet refine. Therefore, we have not observed any error accumulation during IsoNet refine (We added a new paragraph to discuss this in lines 173-179 in the revised manuscript with tracked changes).

2. The benchmarking is done on a simulated dataset. I suggest using/adding a real single-particle dataset, for example, EMPIAR 10045, the benchmark dataset for Relion, to do the benchmarking. The authors could compare the missing wedge effect at different particle orientations, their corrected structures, and the structures recovered from averaging.

Response: Thank you for your suggestion! We now have added another figure showing the result of IsoNet for EMPIAR 10045. The ribosome can be directly segmented in 3D using a threshold in the IsoNet corrected tomograms. The atomic models of ribosomes fitted well into the density of IsoNet corrected tomograms. We added a figure and a paragraph to illustrate this (New Fig. 2 and lines 240-250)

3. The philosophical discussion in the Discussion section is very meaningful. However, this leads to a question of whether IsoNet can do a good job with highly diverse structures. I noticed all three real datasets contain relatively simple structures with many repetitive structural patterns. But a lot of tomograms contain very crowded and diverse subcellular structures. Therefore, it would be helpful if the authors can find a subcellular dataset with diverse structures in a crowded cytoplasm environment to show IsoNet's performance.

Response: We now have shown the IsoNet performance in multiple cellular tomograms in the revised manuscript (Figs. 5-7). We have discussed the performance of synaptic vesicles, ER, small proteins, vesicle tethers, microtubules, microtubule luminal proteins in the neuron tomograms (Fig. 5 and lines 334-349), confirming IsoNet performs well on diverse structures in a crowded cytoplasm environment. We have included more examples of cellular tomograms in the revised manuscripts and a new software tutorial for using IsoNet

for cellular samples with IsoNet (New Fig. 7 and lines 378-393).

4. About the application of IsoNet, facilitating particle picking is definitely an important one. For example, using Laplacian of Gaussian as mentioned by the authors. I think the authors can do a simple experiment on a single particle dataset, for example, EMPIAR 10045, by comparing Laplacian of Gaussian particle picking on tomograms with and without IsoNet correction. The results can be evaluated by whether subtomogram averaging improves from the picked particles.

Response: For the revised manuscript, we have processed the EMPIAR-10045 with IsoNet (New Fig. 2 and lines 240-250). IsoNet greatly enhanced the contrast of the ribosome tomograms, particularly in the X-Z and Y-Z planes, thus enabling 3D particle picking. We note, because ribosome particles are exceptionally large, they can be picked in the x-y plane prior to IsoNet processing, even though not so for the other two planes, see new Fig. 2a. For particles of typical sizes, particle picking even in the x-y plane is often problematic. Indeed, enhanced particle-picking capability has been documented by IsoNet users (see recent publication <https://pubs.rsc.org/en/content/articlepdf/2022/fd/d2fd00022a>).

5. Missing discussion of time cost: training time and prediction on tomograms. Also, does each iteration contain one epoch training?

Response: It takes about 10 hours for 30 iterations of IsoNet refine with the default network structure, using 4 Nvidia-1080Ti. This is the standard timing we elaborated in the tutorial file accompanied by our software package. The time consumption to *Predict* step using 4 Nvidia-1080Ti for one tomogram of 1000x100x300 pixels is about 5 minutes. Each iteration contains 10 epochs. We included these details in the result and method sections of the revised paper (Lines 169-170, lines 203-204, lines 712-713 in the revised manuscript and in IsoNet tutorial https://github.com/Heng-Z/IsoNet/blob/master/IsoNet_v0.2_Tutorial.pdf).

Minor concerns:

1. Some grammar errors: for example Ln 89 'In the cryo-EM field', Ln 91 'the missing wedge information'

Response: We have changed both (see new lines 69-70 and 75-76)

2. No previous deep learning works for cryo-ET tasks are discussed.

Response: We have added a couple of sentences in the discussions in the revised manuscript. (Lines 71-75)

3. Ln 349-354, these are trivial details that can be moved to the Results or Methods section.

Response: We have moved those to the Results sections in the revised manuscript. (Lines 188-196)

4. Ln 775, please have a citation for overlap-tile strategy.

Response: We have added the citation for overlap-tile strategy (Line 734).

5. Figure 1 caption missing e.

Response: Fixed.

Reviewer #2 (Remarks to the Author):

In this paper, Liu et al present an AI-based approach for reducing anisotropy of cryo-tomographic reconstructions. The isonet program aims to fill the 'missing wedge' by extracting subtomograms at random points in the tomograms, and comparing them in a number of rotations to detect the common information, thereby deriving the missing information from mutually similar objects in different orientations.

The software can be run on the command line or through a gui, and includes steps of deconvolution (similar to warp, producing high SNR tomograms), masking (identifying regions of the tomograms likely to contain relevant density), subtomogram extraction, training, and predicting.

The results presented in the paper are impressive, suggesting this approach is more powerful than others previously proposed.

We've actually tried isonet on our own tomograms of reconstituted COPII vesicles and were very impressed with the results, with membranes becoming visible along the z direction, and many features becoming better resolved and directly interpretable. In our isonet-treated tomograms we have been able to visualise individual COPII inner coat complexes on the surface of vesicles which we were unable to do before, no matter what filter we tried. Moreover, with isonet we were able to segment fully closed membranes semi-automatically by simply setting a threshold in chimera, which is allowing us to streamline the particle-picking process significantly.

However, we have tried the software on cellular tomograms, which are much more heterogeneous, and could not see a significant improvement. The same was reported by other colleagues working with in situ data. This differs from the results presented in the paper where clearly an effect was seen for cellular data. As far as I understand, Isonet works by identifying groups of subtomograms that contain similar features, such as membranes, so is it possible that in an environment that is too crowded with heterogeneous features the performance is lower?

Response: IsoNet recovers missing information by identifying multiscale features in the tomograms. Therefore, existence of objects of similar material property (not necessarily same structures) would suffice for IsoNet to work. For example, the horizontally orientated carbon layer (in our new Fig. 2) can be recovered, even though there is no carbon layer at vertical orientation. This indicated that IsoNet can learn information of one object (e.g., a molecule or carbon layer) from all other objects, because all complex objects can be decomposed into similar dots and lines (i.e., similar property). Thus, the crowded and heterogeneous cellular environment may enable better generalization capacity of the network, given enough training dataset.

The observed lower performance of cellular tomograms could be due to their low signal-to-noise ratio. In this revised version, we have incorporated a paragraph to explain how to optimize parameters for cellular subtomograms (new Fig.7, Lines 378-393). With those parameters tuned, we are able to routinely achieve considerable performance for cellular tomograms.

I am happy to recommend the paper for publication, but I would like to see a detailed description of the protocol and parameters used to optimise the in-cell data, with a discussion of potential difficulties related to heterogeneous samples. Could the authors suggest optimum settings for 'clean' purified samples and heterogeneous in situ data (or provide details of the settings used for each individual dataset)?

Response: We now have prepared a new tutorial for cellular tomograms, and now have included details to optimize IsoNet settings in both results and methods sections (lines 378-393 and lines 829-838), for 'clean' purified samples and heterogeneous in situ data. (This document is now provided as part of https://github.com/Heng-Z/IsoNet/blob/master/IsoNet_v0.2_Tutorial.pdf and included as part of the additional review materials in this revision).

Other comments:

1. We played quite a bit with the masking step but did not succeed in masking out the carbon. We had to trim the volume to include the central portion only, use the trimmed tomogram for the training set, and then predict on the full tomograms. It would be useful if in addition to crop_z, also a crop_x and crop_Y options were added to mask out peripheral areas, so these manual cropping steps can be avoided.

Response: Thank you for your suggestion! We now have added another functionality to IsoNet whereby user can draw polygons to define the area of the sample. The carbon areas can be masked out with this function. The detailed usage of the polygon mask is in the new version of the IsoNet (https://github.com/Heng-Z/IsoNet/blob/v0.2alpha/tutorial/IsoNet_v0.2_Tutorial.md).

2. In the introduction, other methods of recovering the missing wedge are dismissed: "However, such assumptions have limited information content (or "entropy") and may not always hold true, given the complexity of biological systems.". The authors should explain with more detail why these methods have limited information content, and how IsoNet does not suffer from the same shortcomings, either in the introduction or discussion.

Response: We have provided explanations for both by providing examples for each. IsoNet does not suffer from the same limitation because no such assumptions are made. We have added the explanation in lines 51-55 of the revised manuscript.

3. It should be stated in the abstract and at the end of the introduction that IsoNet works for tomograms at a pixel size of ~ 10 Å and above to avoid confusion that it can be used on high-resolution tomograms.

Response: We have stated at the end of introduction that IsoNet works on low resolution tomograms with pixel size of ~ 10 Å to avoid confusion (lines 78-79).

4. Can the authors discuss whether there could be overfitting/confusion if similar molecules are present in the tomogram? Could information from different but similar molecules be 'mixed' to fill in missing wedges of similar ones?

Response: Although overfitting should always be considered when dealing with noisy data, such as cryoET data. We believe that IsoNet won't be confused between different but similar molecules to cause overfitting. This is because missing information recovery is performed across multiple scales in the IsoNet neural network, which begins with detecting features in a very local area (e. g. 3x3x3 pixels patches) towards a very large (e.g., 64x64x64 pixels). At the small scale, even the slightest *differences* among *similar* molecules would be readily distinguished by the network. Also, the network does not simply recognize *molecules* per se, rather, it would detect small components of the *molecules*. IsoNet algorithm penalizes artificially adding small component to a molecule (thus leading to the type of overfitting as suggested).

In light of this concern, we have added this point and mentioned another way to alleviate possible overfitting by adding more diverse data for training (see lines 188-196 and lines 440-446 in the revised manuscript), in addition to the three ways already described in the original manuscript (lines 188-196).

5. Denoising: will the simulated backprojected noise be a good model independently of the reconstruction algorithm (i.e. sirt versus backprojection)?

Response: No, the choice of noise models depends on the reconstruction algorithm. In the new version of IsoNet, we have provided three noise models for various tomogram reconstruction algorithms: simple backprojection, backprojection with ramp filter, and backprojection with hamming filter. These noise models correspond to three tomograms reconstruction methods in IMOD (<https://bio3d.colorado.edu/imod/doc/tomoguide.html>): SIRT, weighted back projection, and reconstruction with hamming-like filter, respectively. User can choose different noise reconstruction methods for their tomograms accordingly. We wrote this in the revised methods sections (lines 669-674).

Minor Points:

1. Line 636: contrast inverted with respect to what? i.e. are users expected to start with black density and will the starting contrast affect the result?

Response: We have specified that the contrast inverted tomograms as write density on black background in the revised manuscript (lines 569-570). IsoNet starts with black density (default in IMOD), which is inverted to white during IsoNet mask creation, refine and predict. The program automatically inverts the reconstructed density back to black to produce results. Nonetheless, it does not matter whether the starting tomogram density is black or white, IsoNet also revert back to the original density convention, so users don't need to be concerned of.

2. When describing the refine 1 step details, it needs to be specified that the MW is applied along the zy axis of the unrotated tomograms. As it reads currently, I was under the impression the MW is applied along the same direction as the 'true' MW, i.e. rotated together with the tomogram, which doesn't make sense. Fig 1c, top panel, clarified that for me but it needs to be written clearly too.

Response: We have rewritten these sentences and made those clear. (Lines 147-148 and lines 460-461 in the revised manuscript)

3. "Its application to high-resolution cellular tomograms should also help identify differently oriented complexes of the same kind for near-atomic resolution sub-tomogram averaging." – Remove 'near-atomic' as this claim is substantial and requires proof.

Response: We have removed "near-atomic" in the revised manuscript. (Line 15)

4. The paper contains grammatical errors and imprecisions, please re-read the paper and correct them. Some are listed below, but there were too many to comprehensively correct all of them. Also, references are missing, please see below for details.

Response: Thank you for pointing it out. We re-read the paper and corrected the grammar and citation problems.

- "Examples abound, ranging from pleomorphic viruses, to cellular organelles, to large-scale cellular structures like synapses between neurons." – Cite a range of papers here.

Response: We now have added citations in the revised manuscript (Lines 22-24)

- "Many viruses, notably those involved in devastating pandemics such as SARS-CoV-2, influenza viruses, and human immunodeficiency viruses (HIV), are pleomorphic in the organizations of their proteins and genomes." – Again, cite references.

Response: We have cited the references. (Lines 24-25)

- "This technique requires collecting a series images of the sample at different tilt angles, called "tilt series"." – Should be: 'called a "tilt series"'.
Should be: 'called a "tilt series"'.
Response: Thank you. We now have changed the manuscript as you suggested in the revised version. (Line 36)

Response: Thank you. We now have changed the manuscript as you suggested in the revised version. (Line 36)

- "low signal-to-noise ratio (SNR) for the cryo tomogram." – Should be 'in the cryo tomogram'.

Response: We now have replaced "for the cryo tomogram" with "in the cryo tomogram" (Line 38)

- The paragraph beginning: "To reveal such molecular sociology across viruses or inside cells, cryogenic electron tomography (cryoET) has become the tool of choice." Should contain more references, see cryo-ET reviews (e.g. Saibil & Orlova, Wan & Briggs).

Response: Thank you. We have cited these papers in the revised manuscript. (Lines 34-35)

- "the tilt range for cryoET is usually restricted to about $\pm 70^\circ$." - 60° is the more commonly used tilt range.

Response: Agree. We have replaced it to 60° in the revised manuscript. (Line 40)

- "to constraint the structural features in reconstructed tomograms." – Should be "to constrain".

Response: Yes. We have modified the manuscript accordingly. (Line 52)

- "In the field of computer vision, convolutional neural network (CNN) has been applied to various tasks, such as object recognition, image segmentation, and classification, often achieving high performance" – Again, cite references.

Response: We now have cited papers in the revised manuscript. (Lines 68-69)

- "Here, we have developed a CNN-based software system, called IsoNet, for isotropic reconstruction of electron tomogram." – Should be 'reconstruction of tomograms'.

Response: We have modified the manuscript as suggested. (Line 78)

- "The resolution at Z-axis reaches about 30\AA resolution as measured by the gold-standard Fourier shell correlation (FSC) criterion." – Replace 'at the Z-axis' with 'within the missing-wedge'.

Response: We have replaced 'at the Z-axis' with 'within the missing-wedge' in the revised manuscript. (Line 81)

- "thousands of sub-tomogram particles for future near-atomic resolution cryoET studies." – Replace 'near-atomic' with 'high resolution'.

Response: We have replaced 'near-atomic' with 'high resolution' in the revised manuscript. (Line 89)

- "Thus, it is possible to recover the missing information by merging information from similar features present in the same tomograms but at different orientations relative to each other." – Remove 'thus'.

Response: We have removed 'thus' in the revised manuscript. (Line 94)

- "Among the 5 steps, Refine and Predict relies on graphical processing unit (GPU) that provides superior processing power." - Should be: "Among the 5 steps, Refine and Predict requires GPU acceleration".

Response: Thank you! We have modified the text as you pointed out. (Line 104)

- "particularly on Z-axis." – Should be 'particularly in the Z-axis'.

Response: We have replaced 'particularly on Z-axis' with 'particularly in the Z-axis'. (Line 139)

- "To further improve miss-wedge correction" – Should be: 'missing-wedge'.

Response: We have modified the text as you pointed out in the revised version. (Line 162)

- “ribosome as the second test due to its asymmetric shape and primarily nucleic acid content.”. The ribosome is not formed primarily of nucleic acids, remove this part.

Response: We have removed the mentioned part in the revised manuscript. (Lines 216-217)

- “Importantly, our isotropic 3D reconstruction shows that the quality of the structure is similar across all directions, allowing biological structures to be interpreted adequately (Fig. 2c and Supplementary Video 1).” – This is true for low resolution features only and this should be clarified.

Response: We have explicitly added “at low resolution” in this sentence in the revised manuscript. (Line 271)

- “Next, we tested the performance of IsoNet on resolving structures within cellular organelles by processing tomograms of flagella of Trypanosoma. Brucei using IsoNet.” – Please cite the paper where these tomograms were produced here and remove the ‘.’ After Trypanosoma.

Response: We now have cited the paper and removed the dot in the revised manuscript. (Line 294)

- Correct all instances of ‘Weiner’ to ‘Wiener’

Response: We have replaced all ‘Weiner’ to ‘Wiener’ in the revised manuscript. (Lines 128 and 570)

REVIEWER COMMENTS

Reviewer #1 (Remarks to the Author):

The authors addressed all my concerns. I only have one additional suggestion to emphasize the significance and innovation of the proposed work:

There are many deep learning based methods proposed in cryo-ET. I would like to see a comprehensive discussion on how they deal with missing wedge effects. If a method does not consider missing wedge effects, what are the potential errors that may occur? If a method considers missing wedge effects, how does its approach compare with the proposed approach in principle?

I would recommend the manuscript be accepted after minor revision.

Reviewer #2 (Remarks to the Author):

We are happy with the revisions and recommend the manuscript for publication. The manuscript is much improved and useful features have been added to the software. Thank you.

Response to Reviewers' Comments

Summary of responses: We thank the reviewers comments and the constructive suggestion. In this revision, we have added lines 68-79 in the introduction section to incorporate a comprehensive overview of all existing cryoET packages with deep learning methods.

To facilitate your navigation of this document, the reviewers' original comments are pasted in **black**, and our responses are in **blue**.

REVIEWER COMMENTS

Reviewer #1 (Remarks to the Author):

The authors addressed all my concerns. I only have one additional suggestion to emphasize the significance and innovation of the proposed work:

There are many deep learning based methods proposed in cryo-ET. I would like to see a comprehensive discussion on how they deal with missing wedge effects. If a method does not consider missing wedge effects, what are the potential errors that may occur? If a method considers missing wedge effects, how does its approach compare with the proposed approach in principle?

I would recommend the manuscript be accepted after minor revision.

Respond: In the revised manuscript, we add a more comprehensive review of all existing cryo-ET methods incorporating deep-learning strategies (see lines 68-80). These packages use deep-learning in various cryoET related tasks, including particle picking(Moebel et al., 2021; Wagner et al., 2019), classification(Che et al., 2018; Xu et al., 2017), segmentation(Chen et al., 2017; Xu et al., 2019) and denoising(Bepler, Kelley, Noble, & Berger, 2020; Buchholz et al., 2019; Tegunov & Cramer, 2019), but not the missing wedge problem. We now indicate in this paragraph that none of these existing methods successfully dealt with the intrinsic missing-wedge problem of cryoET reconstruction and artefacts arising from it. In the Discussion paragraph, we noted that IsoNet and sub-tomogram averaging compensate for the missing-wedge problem through the same principle (lines 407-408). But IsoNet does so automatically.

Reviewer #2 (Remarks to the Author):

We are happy with the revisions and recommend the manuscript for publication. The manuscript is much improved and useful features have been added to the software. Thank you.

References for rebuttal:

- Bepler, T., Kelley, K., Noble, A. J., & Berger, B. (2020). Topaz-Denoise: general deep denoising models for cryoEM and cryoET. *Nat Commun*, *11*(1), 5208. doi:10.1038/s41467-020-18952-1
- Buchholz, T. O., Krull, A., Shahidi, R., Pigino, G., Jekely, G., & Jug, F. (2019). Content-aware image restoration for electron microscopy. *Methods Cell Biol*, *152*, 277-289. doi:10.1016/bs.mcb.2019.05.001
- Che, C. Q., Lin, R. G., Zeng, X. R., Elmaaroufi, K., Galeotti, J., & Xu, M. (2018). Improved deep learning-based macromolecules structure classification from electron cryo-tomograms. *Machine Vision and Applications*, *29*(8), 1227-1236. doi:10.1007/s00138-018-0949-4
- Chen, M., Dai, W., Sun, S. Y., Jonasch, D., He, C. Y., Schmid, M. F., . . . Ludtke, S. J. (2017). Convolutional neural networks for automated annotation of cellular cryo-electron tomograms. *Nat Methods*, *14*(10), 983-985. doi:10.1038/nmeth.4405
- Moebel, E., Martinez-Sanchez, A., Lamm, L., Righetto, R. D., Wietrzynski, W., Albert, S., . . . Kervrann, C. (2021). Deep learning improves macromolecule identification in 3D cellular cryo-electron tomograms. *Nat Methods*, *18*(11), 1386-1394. doi:10.1038/s41592-021-01275-4
- Tegunov, D., & Cramer, P. (2019). Real-time cryo-electron microscopy data preprocessing with Warp. *Nat Methods*, *16*(11), 1146-1152. doi:10.1038/s41592-019-0580-y
- Wagner, T., Merino, F., Stabrin, M., Moriya, T., Antoni, C., Apelbaum, A., . . . Raunser, S. (2019). SPHIRE-crYOLO is a fast and accurate fully automated particle picker for cryo-EM. *Communications Biology*, *2*. doi:10.1038/s42003-019-0437-z
- Xu, M., Chai, X., Muthakana, H., Liang, X., Yang, G., Zeev-Ben-Mordehai, T., & Xing, E. P. (2017). Deep learning-based subdivision approach for large scale macromolecules structure recovery from electron cryo tomograms. *Bioinformatics*, *33*(14), i13-i22. doi:10.1093/bioinformatics/btx230
- Xu, M., Singla, J., Tocheva, E. I., Chang, Y. W., Stevens, R. C., Jensen, G. J., & Alber, F. (2019). De Novo Structural Pattern Mining in Cellular Electron Cryotomograms. *Structure*, *27*(4), 679-691 e614. doi:10.1016/j.str.2019.01.005